

# Wuhan MST radar: Technical features and Validation of wind observations

Lei Qiao[1,2], Gang Chen[2], Shaodong Zhang[2], Qi Yao[3], Wanlin Gong[2], Mingkun Su[1],
Feilong Chen[4], Erxiao Liu[1], Weifan Zhang[2], Huangyuan Zeng[2], Xuesi Cai[1], Huina Song[1],

Huan Zhang[1], Liangliang Zhang[1]

[1] Communication Engineering School, Hangzhou Dianzi University, Hangzhou 310018, China
[2] Electronic Information School, Wuhan University, Wuhan 430072, China
[3] Nanjing Research Institute of Electronic Technology, Nanjin 210013, China
[4] Information Engineering School, Nanchang Hangkong University, Nanchang 330063, China

*Correspondence to*: Gang Chen (g.chen@whu.edu.cn)

**Abstract.** The Wuhan MST radar is a 53.8 MHz monostatic Doppler radar, located in Chongyang, Hubei Province, China, which has the capability to observe the dynamics of the mesosphere-stratosphere-troposphere region in the subtropical latitudes. The system is composed of 576 Yagi antennas with square distribution, and the maximum peak power is 192 kW. The Wuhan MST radar is efficient and cheap, which applies simplifier and more flexible architecture. It includes 24 big TR

modules, and the row/column data port of each big TR module connects 24 small TR modules via the corresponding row/column feeding network. Each antenna is driven by a small TR module with peak output power of 300 W. The arrangement of the antenna field, the functions of the timing signals, the structure of the TR modules, and the clutter suppression procedure are described in detail in this manuscript. We compared the MST radar observation results with other instruments and related models in the whole MST region for validation. Firstly, we made a comparison of the Wuhan MST

radar observed horizontal winds in the troposphere and low stratosphere with the radiosonde in the short term, as well as the ERA-interim data sets (2016 and 2017) in the long term. Then, we made a comparison of the observed horizontal winds in the mesosphere with the meteor radar and the HWM-07 model in the same way. In general, good agreements can be obtained, and it indicates that the Wuhan MST is an effective tool to measure the three-dimensional wind fields of the MST region in the short-term and long-term.

## 1 Introduction

The mesosphere-stratosphere-troposphere (MST) radar has been used for observing the dynamics of the lower and middle atmosphere up to 100 km altitude for several decades (Hocking et al., 2011), since Woodman and Guillen observed radar echoes from the stratospheric and mesospheric heights with the Jicamarca radar in 1970s (Woodman et al., 1974). In general, large antenna array is applied by these MST radars to measure the weak echoes scattered by the turbulences (Green et al.,

1979). Many MST radars have been developed world-wide by different countries and groups, and the MST community plays





a significant role. According to the antenna array shape, the existing MST radar in the world can be divided into two types: the square array arranging the elements in a square gird and the circular array arranging the elements in a triangular grid. The MST radars using the square array mainly include the Jicamarca radar (Woodman et al., 1974), the Sousy radar (Schmidt et al., 1979), the Poker Flat radar (Balsley et al., 1980), the Esrange MST radar (Chilson et al., 1999), the Gadanki radar (Rao et al., 1995), the Chung-Li radar (Rottger et al., 1990) and the NERC MST radar (Vaughan, 2002). The MST radars using the circular array include the MU radar (Fukao et al., 1985), the EAR radar (Fukao et al., 2003), the MAARSY radar (Latteck et al., 2012) and the PANSY radar (Sato et al., 2014).

Because of the expense, the MST radar has been developing slowly in Chinese sector. Until 2008, the Wuhan MST radar and Beijing MST radar began to construct with the support of Meridian Project of China (Wang, 2010). We have introduced the two MST radars of Chinese Meridian project in 2016 (Chen et al., 2016). This manuscript briefly introduced the antenna array of the Beijing and Wuhan MST radars and their preliminary observations. The two MST radars work more than 280 days every year and their data can be freely accessed in the data center for the Meridian Project (http://159.226.22.74/). Thus, the radar system and their data are gained extensive attention and we have received many letters inquiring about the details of the radio system, as well as the data format and reliability. Therefore, we plan to wright a new article to response the readers and users, who want to build a low-cost MST radar or apply the data of the MST radars of Chinese Meridian project. The manuscript presents more details of the Wuhan MST radar including the photos of some important circuits, as well as its recorded data.

The Wuhan MST radar is located in Chongyang, Hubei Province, China (29.5°N, 114.1°E). The location is away from the bustling city, so as to better avoid interference of radar echoes by radio noise. Considering this is China's first attempt to develop its own MST radar, the radar station is not selected in some areas of great difficulty in construction, such as equatorial low latitudes, polar regions and plateaus. Chongyang is located in the central plain of China, which is an appropriate choice. As one of few MST radars in the midlatitudes, it can be one important member of the global MST radars. In addition to the scientific research goals, the Chongyang station also serves as a students' training base for the practice of radar technologies and meteorological applications. The Wuhan MST radar was completed preliminarily in 2011. The system was upgraded in 2016, and the TR modules were updated for better stability and better detection capability. The facility costs only about $1,000,000.00, which is far lower than the high cost of other MST radars. However, it provides an average power aperture product (PAP) product of $3.2 \times 10^8$ $Wm^2$. Considering the balance of system performance and project implementation, simpler and more flexible architectures are applied in the system.

The first aim of the present manuscript is to introduce the technical features of the Wuhan MST radar. In particular, the antenna field, the timing signal, the TR module, the digital receiver and the clutter suppression will be discussed in detail. The second aim of this manuscript is to present the the recorded data and compared with the wind fields recorded by other instruments and related models for validation.



## 2 Technical features

**Figure 1.** Schematic block diagram of the Wuhan MST radar located in Wuhan, China. The small TR module controllers, the small TR modules, the feeding network and the antenna array are installed in the antenna field. Other modules are installed in the observation house.

The Wuhan MST radar is arranged in a 24 × 24 matrix with a side length of 96 m, which consists of 576 Yagi antennas. Each antenna is driven by an individual small TR module (300 W). According to the antenna radiation pattern, the beam width is 3.2°. The shortest width of the subpulse width is 1 μs to satisfy the requirement for a maximum range resolution of 150 m. The system allows very high flexibility of waveform parameter for different detection modes (low mode, middle mode, and high mode). The hardware of the Wuhan MST radar consists mainly of five subsystems: the antenna array, the TR module, the radar controller, the digital transceiver, and the signal processor. Fig. 1 shows the schematic block diagram of the system.



The signal scattered by the turbulence is received by the antenna array, then sent to the TR module by the feeding network.
The TR module includes 24 big TR modules installed in the observation house and 576 small TR modules installed in the shelters. The big TR controller receives the timing signal from the timing generator in the radar controller by parallel port. The big TR controller converts the timing signal into multiplex signals to control the 24 big TR modules. At the same time, the BIT-TR signal is sent to the main controller to monitor the condition of the big TR module. The big TR controller also handles communication with 36 small TR module controllers over twisted-pair, and each big TR module controller
corresponds to 16 small TR modules. The row data port in the big TR module connects 24 small TR modules in a row of the antenna array via the row feeding network, while the column data port connects 24 small TR modules in a column via the column feeding network.

The radar controller consists of a master oscillator, a frequency synthesizer and a control timing generator and a main controller. The master oscillator generates the reference clock of 100 MHz. Then, the frequency synthesizer generates the
sampling clock of 80 MHz, the direct digital synthesizer (DDS) clock of 240 MHz, and the control reference clock of 20 MHz. Combined with the reference clock of 20 MHz and the commands from the main controller, the control timing generator generates various timing signals for radar control.

The digital transceiver consists of the DDS module, and the 24-channel receiver. The DDS module is used to generate the binary-phase-coded continuous wave for transmitting and the test signal for channel calibration. The 24-channel receiver
includes the amplitude limiter, the bandpass filter, the linear amplifier, the analog-to-digital converter (ADC) and the digital down converter (DDC). The amplitude and phase weight algorithm is realized in the digital beam forming (DBF) module.

The data processing implemented in the digital signal processing (DSP) chip involves pulse compression, coherent averaging, fast Fourier transform (FFT), and spectrum averaging. Then, the output of the DSP chip is transferred to the online data processor by a peripheral component interconnect (PCI) bus. The main functions of the online data processor are
clutter suppression, parameter extraction, wind field retrieval and wind field displaying. Eventually, the product data of the wind filed in the troposphere, lower stratosphere, and mesosphere is produced.

## 2.1 Antenna field

Fig. 2 shows the arrangement of the Wuhan MST radar antenna field. The Yagi antennas are arrayed on grids of squares about 4 m on a side, and this element spacing allows no grating lobe beam scanning up to an angle of about 20° from zenith.
The voltage standing wave radio (VSWR) of the antenna is less than 2.5.

As shown in the right part of Fig. 2 for e.g. S0101, there are 144 shelters mounted at regular intervals, and each one consists of 4 small TR modules, a 4:1 row divider/combiner unit (DCU), a 4:1 column DCU, and a power supplier and a small TR controller. It should be pointed out that the 36 small TR controllers are located in the shelters of odd rows and columns. The eight yellow boxes, labeled as F1-F8, represent the row feeding boxes (F1-F4) and the column feeding boxes
(F5-F8). Each feeding boxes contains six 6:1 DCUs, and each one feeds four 4:1 DCUs in the shelters. The DCUs in the row/column feeding boxes are all fed by the big TR modules in the observation house, and the row or column drive state is



switched by the control signal. The row/column data from the 24 big TR module feeding the 6:1 row/column DCUs is labeled as R1-R24/C1-C24.

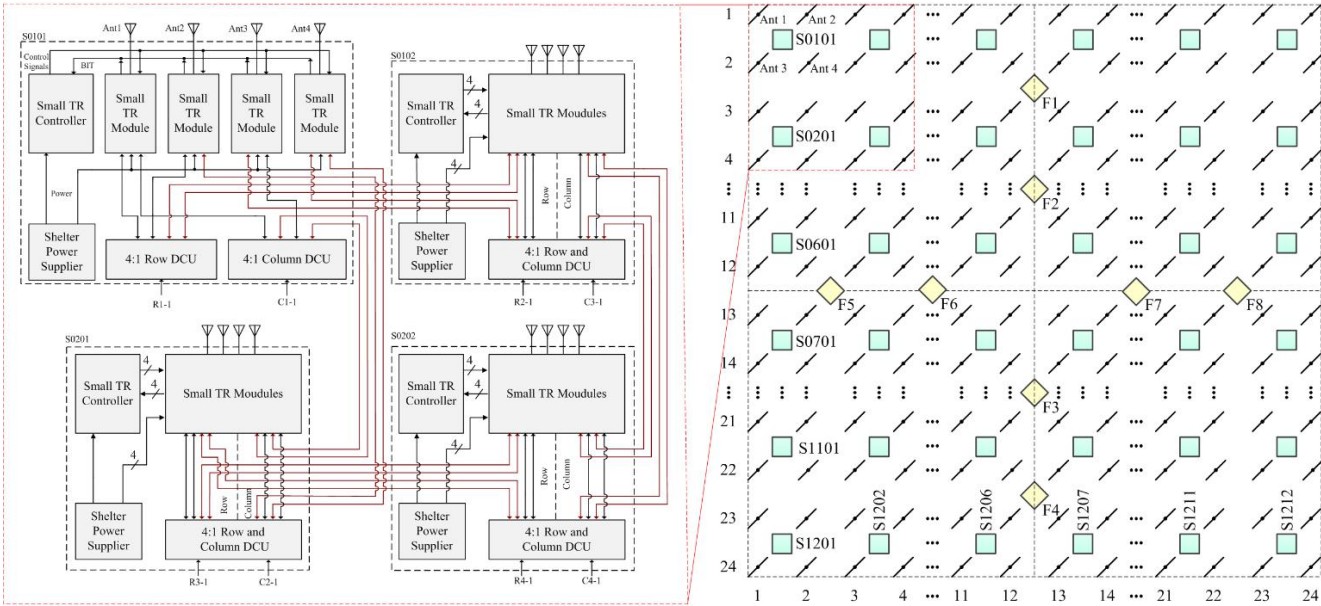

**Figure 2.** Arrangement of the Wuhan MST radar antenna field and inter connections of four surrounding shelters.

As illustrated by the red box in the right part of Fig. 2, there are four shelters (S0101, S0102, S0201, S0202) and the surrounding antennas. The left part of Fig. 2 shows the inter connections of the shelters, and the lines of different shelters are red for easy review. In the shelter S0101, the row DCU is fed by R1-1, which respects the first divided signal of R1. The

other 4 ports of the row DCU connect to the row data pots of small TR module 1 and 2 in S0101 and S0102 respectively. Similarly, the column DCU is fed by C1-1, and the other 4 ports connect to small TR module 1 and 3 in S0101 and S0201 respectively. By that analogy, the row/column DCUs of the shelter S0102, S0201, and S0202 connect to proper data ports of the small TR modules. With this system configuration, the beam can be steered to north-south direction in the row drive state and east-west direction in the column direction. The antennas are aligned in the northwest-southeast direction for

symmetrical radiation pattern. The beams are usually steered to five directions (vertical, north, south, east, west) with off-zenith angles of 15°.

The feeding network of the Wuhan MST radar uses feeding cables of equal length. In this situation, the feeding cables of different channels have stable characteristics, which need no compensation. The big TR modules, the 6:1 dividers and combiners, the 4:1 dividers and combiners , and the antennas are connected via coaxial cable (-3dB/100m). The feeding

cables of above modules are 100 m, 50 m, 10 m, and 7m respectively. Therefore, the feeding line loss from the TR module in the observation house to the end antenna of the array is about 5 dB.





## 2.2 Timing signal

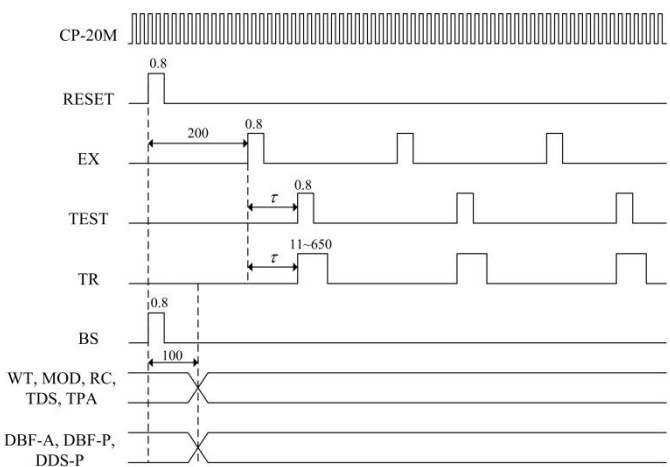

**Figure 3.** Timing diagram of the signals for radar operation.


**Table 1.** Description of control signals.

| Signal | Function | Description |
|--------|----------|-------------|
| WT | Work/Test | 0: Test<br>1: Work |
| MOD | Mode control | 000~111: Low mode 1, Low mode 2, Low mode 3, Middle mode 1, Middle mode 2, High mode 1, High mode 2, High mode 3 |
| RC | Row/Column switch | 0: Row<br>1: Column |
| TDS | Test signal Doppler shift | 000~111: 0, 1, 2, 5, 10, 20, 50, 100 Hz |
| TPA | Test signal power attenuation | 000~1111: 7, 15, 23, 31, 38, 46, 54, 62 dB |
| DBF-A | DBF amplitude weighting coefficient | 0000~1111: 32 sets of amplitude weighting coefficient |
| DBF-P | DBF phase weighting coefficient | 00000~11000: 41 sets of DBF phase weighting coefficient(-20°~20°) |
| DDS-P | DDS phase weighting coefficient | 00000~11000: 41 sets of DDS phase weighting coefficient(-20°~20°) |

All timing signals for radar operation are generated by the timing generator in the radar controller. Fig. 3 presents the timing diagram of the signals at the radar controller. They are generated from the reference clock signal of CP-20M. The RESET signal is used to activate the timing generator. The excitation (EX) signal is used to generate the pulse repetition period



(PRP), and the value is different according to different detection modes. The EX signal is 200 μs later than the RESET signal. The sign τ is the delay of the TEST signal and the TR signal compared to the EX signal. The delay can be adjusted by software, and the rang is from half period of clock ahead to half period of clock delay. The transmitted pulse width of the TR signal is from 11 μs to 650 μs, which is related to the compressed pulse width and the coding scheme. The beam switch (BS) signal controls the switching of the vertical beam and four oblique beams, and it is synchronised with the RESET signal.

Besides the transmitting and receiving timing signals, the radar controller also generates the control signals for system control, which are shown in the last two lines of Fig. 3. It should be pointed out that the control signals are valid 100 μs after the RESET signal rising edge. The different control signals are listed in Table I. The work/test (WT) signal causes the system to operate in work mode or test mode. The test mode is applied for digital receive channels calibration. The mode control (MOD) signal is used to select the observation mode: low mode (troposphere), middle mode (stratosphere), and high

mode (mesosphere). The low mode 1, middle mode 1, and high mode 1 are usually selected under normal operating conditions. The corresponding Doppler resolution and radar scanning time are 0.53 m/s and 5 min. The parameters of the other observation modes can be set flexibly for applications of high Doppler resolution. The row/column switch (RC) signal is used to control the R/C switch in the small TR modules and big TR modules. The test signal Doppler shift (TDS) signal sets the test signal to establish different value of Doppler shift, which servers the digital channel calibration. The test signal

power attenuation (TPA) signal controls the attenuation coefficient of the test signal, so as to prevent damage to the digital receiver. The DBF amplitude weighting coefficient (DBF-A) signal and the DBF phase weighting coefficient (DBF-P) signal are used for beam forming in the DBF module. The DDS phase weighting coefficient (DDS-P) signal causes the DDS to generate the beams of different zenith angles with a step size of 1°, and the maximum angle is 20°.

**2.3 TR Module**

Block diagram and photograph of the big TR module is shown in Fig. 4. The big TR module amplifies the DDS output (53.8 MHz) supplied from the digital transceiver module, and feeds it to the feeding network. This module consists of a three-stage amplifier with a gain of 40.6 dB. The D2089UK and D1001UK are employed in the first and second power amplifier stage, whose drain-source voltage is 3.2 V and 2.5 V respectively. They are metal gate RF silicon field effect transistors (FET) with different power output. The DDS output (10 dBm) is amplified to 30 dBm in the first stage, while the first stage output is

amplified to 40.8 dBm in the second stage. The push-pull power metal oxide semiconductor (MOS) transistor BLF278 is employed in the final stage with a gain of 9.8 dB, whose drain-source voltage is 24 V. Built-in test (BIT) technique is adopted to detect VSWR and power information of the amplifier output (50.6 dBm) via a directional coupler. The information is not only used to monitor status of the big TR module, but also avoid damage to the amplifier. The insertion loss of the T/R switch and R/C switch in the big TR module is about 0.3 dB. Therefore, the total output becomes 50 dBm.



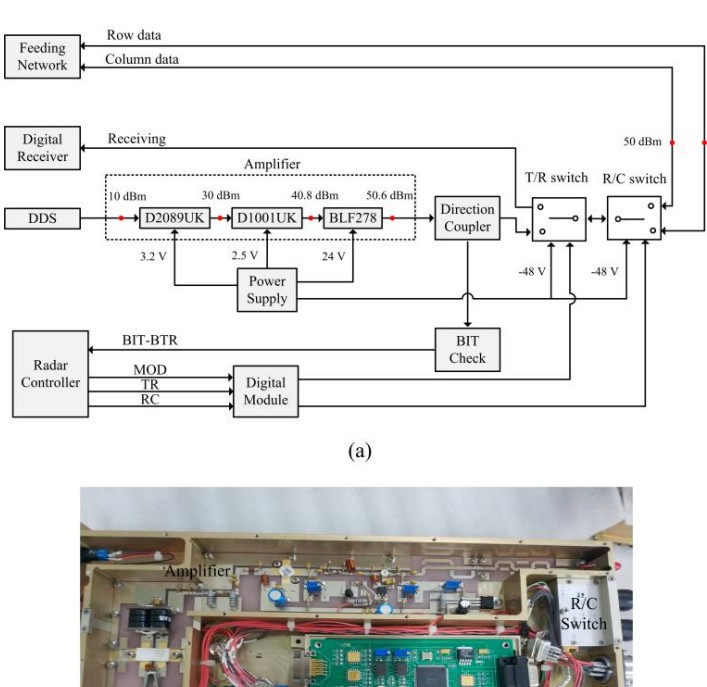

(a)

(b)


**Figure 4.** Block diagram (a) and photograph (b) of the big TR module.

The control signals transferred from the big TR controller are transformed over twisted-pair for better transmission ability. The TR signal (differential signal) is converted into a single ended signal by the differential converter, and then it controls

the R/C switch to realize transformation of row/column. The differential receiver chip DS96F173 is used as the differential converter, which allows operating at high speed while minimizing power consumption. The TR signal controls the T/R switch to receive the signal from the row/column data port, or transmit the amplifier output to the row/column data port. The MOD signal is transferred to the digital module, so as to generate timing signals to control the T/R switch and R/C switch for different observation modes. The recovery time of the two switches is less than 5μs, which reaches the allowable level. The

power supply provides -48 V for the two switches. It should point out that the differential converter is involved in the digital module, as shown in Fig. 4(b).

Block diagram and photograph of the small TR module are shown in Fig. 5. The small TR module consists of various submodules: an amplifier, a T/R switch, a R/C switch, a BIT module, and a differential receiver. Each row/column signal is divided equally into 24 signals in the feeding network. Considering the attenuation of the dividers and cables (100 m cable: -

3 dB; 1:6 divider: -7.78 dB; 50m cable: -1.5 dB; 1:4 divilder: -6.02 dB), the transmitting signal from the big TR module is

reduced to 31.7 dBm. Then the signal is transmitted through a low power R/C switch with 0.3 dB insertion loss. A two-stage amplifier in the small TR module amplified the signal from 31.4 dBm up to 55.4 dBm, and the drain-source voltage of BLF278 is 40V with a gain of 13.2 dB. Then, a band pass filter centered on 53.8 MHz is provided for spurious emission suppression. The T/R switch in the small TR module has an insertion loss of about 0.5 dB and provides an isolation of 60 dB.

Ultimately, the output of the small TR module is about 300 W, and the signal is fed to the Yagi antenna via a 7m coaxial cable. The low-noise amplifier (LNA) in the receiving channel is a 53.8 MHz tuned amplifier with a gain of 28 dB and a bandwidth of 1 MHz.

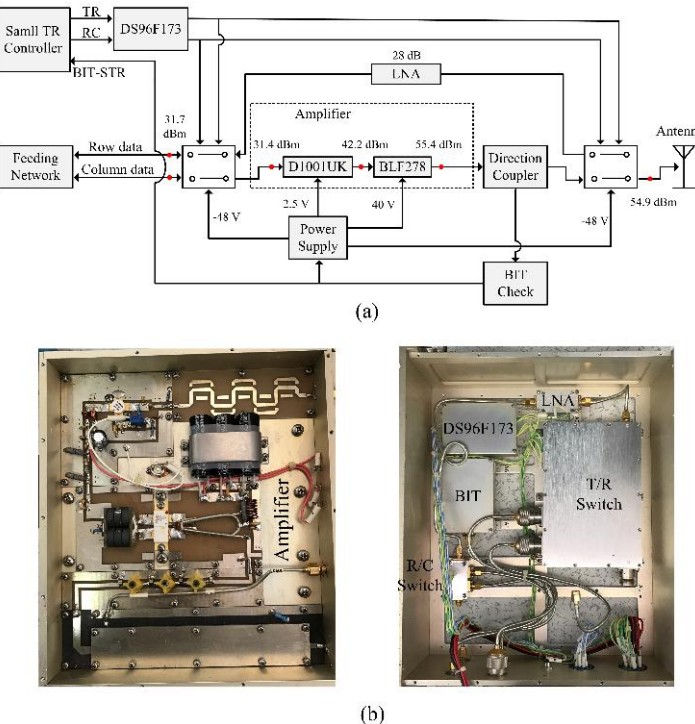

**Figure 5.** Block diagram (a) and photograph (b) of the small TR module.


The two switches embedded in the small TR module allow the controller to select proper signaling pathway, and they are both controlled by two control signals from the small TR radar controller. The small TR module is the easiest damaged part in the system. Therefore, it needs to be repaired every year. As shown in Fig. 5(b), the small TR module adopts the modular design, which is convenient for maintenance.

## 2.4 Digital Transceiver


The digital-up-converter (DUC) chip AD9957 is used in the DDS module, which has 1 Gsps internal clock speed with 18-bit IQ data path and 14-bit digital-to-analog converter (DAC). The 16-bit or 32-bit complementary code with different



pulsewidth (1μs, 4μs and 8μs) are generated by the DDS module, as well as the test signal. The passive calibration algorithm is used for amplitude and phase calibration of the 24 channels in the receiver. The test signal can be set with different value

of Doppler shift, and is divided into 24 channel signal by the divider. The intrinsic amplitude and phase differences among channels can be extracted by comparing the output of each channel. Then, the amplitude and phase calibration factors are stored in the register. Through correction, the receiver has an amplitude consistency of -0.5-0.5 dB and a phase consistency of -2°-2° after calibration.

In the receiver, the signal firstly goes through the amplitude limiter for protection of the receiver, the linear amplifier with

50 dB gain, the bandpass with the center frequency of 53.8 MHz and bandwidth of 1.5 MHz, respectively. Then, the signal is sampled by the LTC2208 with 80 Msps clock rate. It is an ADC with 16-bit resolution, maximum 130 Msps and 100 dB spurious free dynamic range (SFDR). By directly bandpass sampling, the received signal is aliased to 26.2 MHz. The DDC unit is integrated in the FPGA, which is made up of the numerically controlled oscillator (NCO), the cascade integrator comb (CIC) filter, and the finite impulse response (FIR) filter. In general, the frequency output of the NCO is 26.2 MHz, the

bandwidth of the FIR filter is 1.5 MHz, and the total decimation value is 80. Eventually, the in-phase (I) component and orthogonal phase (Q) component with 1 MHz are transformed to the digital beam forming module for further processing.

**2.5 Clutter suppression**

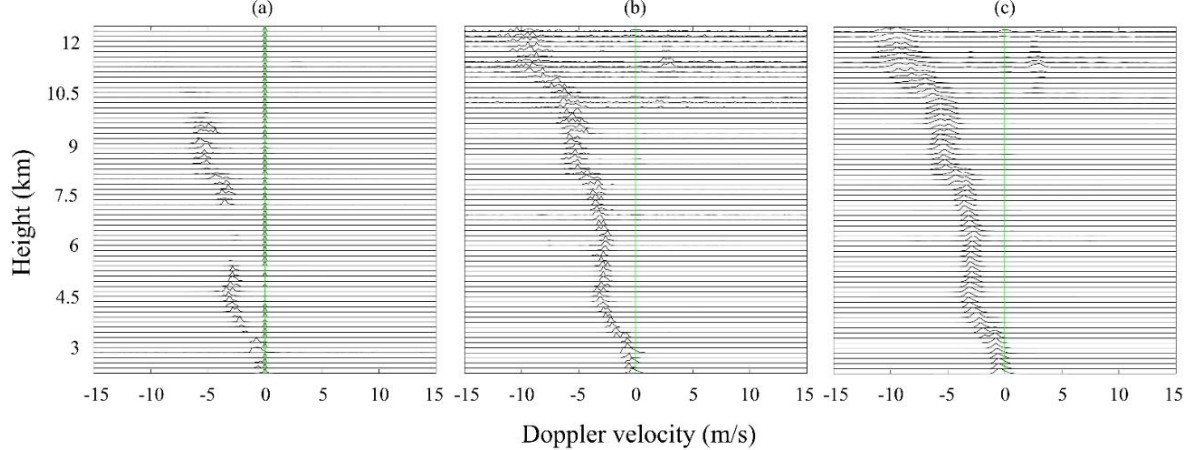

**Figure 6.** Comparison of the original (a), ground clutter suppressed (b), and filtered (c) Doppler spectrum.


The clutter suppression is carried out in the online data processor. Kumar et al. (Kumar et al., 2019) identified the turbulence echo in the multipeaked VHF radar spectra during the precipitation, and here we mainly aimed at ground clutter and high frequency interference in sunny day. Ground clutters from surrounding mountains, trees, and buildings can severely degrade parameter estimates of the turbulence (Schmidt et al., 1979). It is because the weak echoes from the clear air are easy to be

contaminated for the lager amplitude of the ground clutters. From the perspective of radar infrastructure, the construction of



fence is an effective method to isolate ground clutter, i. e. the 10 m height MU radar fence for ground clutter prevention (Rao et al., 2003). The fence is also constructed for the Wuhan MST radar, but there are still some ground clutters in the echoes. Therefore, ground clutter suppression is an essential step for signal processing. The ground clutter echoes have narrow central peak near zero frequency with small temporal changes, and are weakened with increasing altitude.

Fig. 6 shows the processing procedure of the Doppler spectrum recorded by the east beam in the low mode. Note that the Doppler spectrums at the range gates are all normalized. As show in Fig. 6(a), the ground clutters severely bias the desired signals, and most turbulence echoes are submersed. As shown in Fig. 6(b), After the ground clutter suppression, the ground clutters near zero frequency are rejected effectively, and the weak signals appears at the heights of 5.4-7.05 km and 9.75-12 km. The median filter is applied to remove the high frequency interference at each range gate. As shown in Fig. 6(c), the

high frequency interferences decrease, and the power spectrum qualities are improved obviously.

## 3 Validation of wind observations

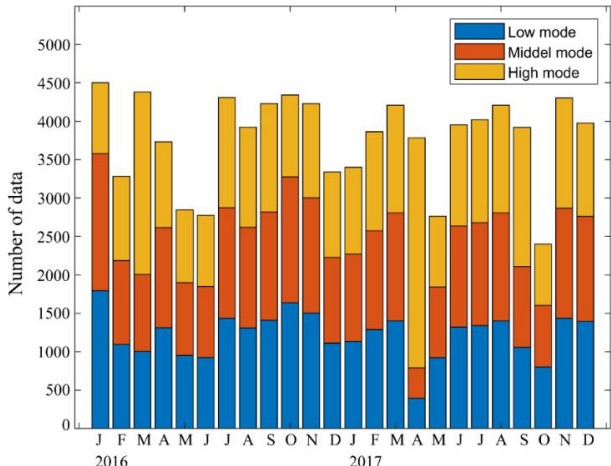

**Figure 7.** The monthly total number of the Wuhan MST radar data in three observation modes during January 2016 to December 2017. Blue for low mode, red for middle mode, and orange for high mode.


The Wuhan MST radar is used for the standard observations of the troposphere, lower stratosphere, and mesosphere in three observation modes: low mode (about 3.5-12 km), middle mode (about 10-25 km) and high mode (about 60-85 km). Height resolution is 150 m for the low mode, 600 m for the middle mode, and 1200 m for the high mode. Under the normal operation, the system usually works in each mode for 5 min in sequence, then takes a break for 15 min. Hence, the wind data

for each mode has 30 min temporal resolution. The Wuhan MST radar is in good running condition during the full-time unattended operation from January 2016 to December 2017. Because of the system failures or external disturbances, severe interference may appear in some data (about 1%). After removing the data with severe interference, the valid data set is present here to demonstrate the performance of the radar. Fig. 7 shows the monthly total number of the Wuhan MST radar





data in low, middle and high modes. According to Fig. 7, the number of the radar data in most months exceeds 3000, except

March 2016, March 2017 and October 2017 during maintenance. On average, the number of daily-mean data is 41 in low

mode, 41 in middle mode, and 44 in high mode. In addition, the Wuhan MST radar took more observations of the

mesosphere in March 2016 and April 2017. Therefore, the data set of the two years provides comprehensive and effective

coverage of the troposphere, stratosphere and mesosphere observations.

## 3.1 Data acquisition

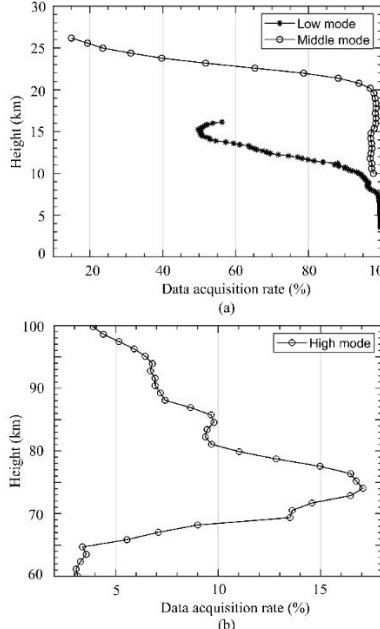

**Figure 8.** The average data acquisition rate from the Wuhan MST radar in low, middle (a) and high (b) observation modes during January
2016 to December 2017.


Data acquisition rate is one of the most important index to describe the MST radar performance, which is calculated using

the relation: 100 × number of samples with valid horizontal wind data / total number of samples (Kumar et al., 2007). The

horizontal wind velocity is estimated by the radial velocity of the four inclined beams through the use of Doppler beam-

swinging (DBS) technique (Anandan et al., 2001). If any one of the inclined beams has serious interference or lower signal-

to-noise (SNR), which led to failure of the horizontal wind velocity inversion, then the samples are judged to be invalid. Fig.

8(a) shows the profile of total data acquisition in the low and middle mode during January 2016 to December 2017. In the

low mode, the data acquisition rate remains >90% at heights of 3.5-10 km, and then decreases rapidly to 50% at height of 15

km. Note that the profile of low mode clearly shows a reversal at heights of 14-16 km corresponded to the tropopause (Chen

et al., 2019). In the middle mode, the data acquisition rate remains >90% at heights of 10-20 km, and then decreases rapidly

to 19% at height of 25 km. Therefore, the connection height of the low mode and middle mode is usually selected at height



of 10 km for optimal data acquisition. In some situations which require high range resolution, the data acquisition rate
(>50%) of the low mode is also available for the heights of 3.5-16 km.

As shown in Fig. 8(b), the data acquisition rate of the high mode is mainly concentrated at heights of 66-86 km with a
maximum up to 17%, which is lower than that of the low and middle mod. It is because the winds in the mesosphere are only
available during the daytime (8 LT-16 LT) in the D region (due to insufficient D region ionization during nighttime) (Rao et
al., 2014). Actually, if the time range is limited in the daytime, the maximum data acquisition of the high mode is more than
50%. The analysis of the data acquisition rate indicates that the Wuhan MST radar can receive the backscattered echoes from
the troposphere, stratosphere and mesosphere effectively.

## 3.2 Tropospheric and low stratospheric observation

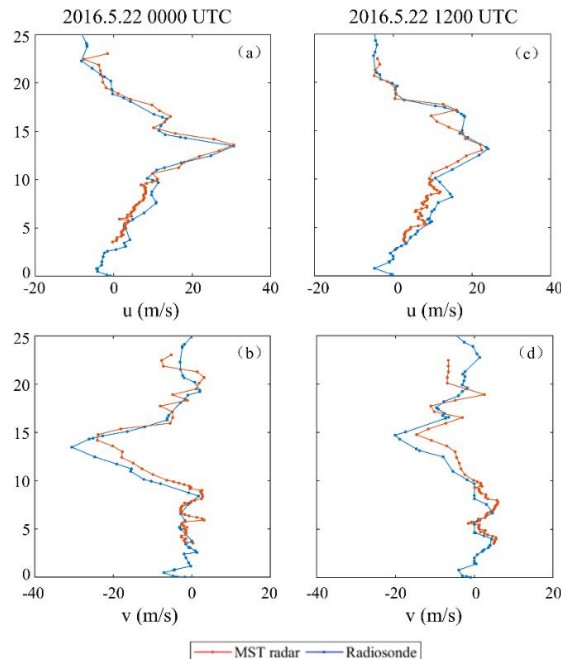

**Figure 9.** The zonal (u) and meridional (v) winds at the heights of 3.5-25 km observed by the Wuhan MST radar (red lines) and the
radiosonde (blue lines) at 00 UT (a, b) and 12 UT (c, d) on 22 May 2016.

In order to verify the validity of the wind measurements in the height ranges of 3.5-25 km, simultaneous observations
obtained from the radiosonde were compared with the Wuhan MST radar observations. The radiosonde launch site (30.6°N,
114.1°E) is about 120 km away from the Wuhan MST radar, and the radiosonde was launched at 00 UT and 12 UT on 22
May 2016. It took about an hour for the balloon to rise up to 25 km, while the repetition period of the Wuhan MST radar is
30 min. Therefore, after the balloon was launched, the following two periods of the Wuhan MST radar data were averaged to
compare with the data from the radiosonde.

Note that since the balloon was detected at regular intervals by the tracking radar, the height resolution of the radiosonde
is not uniform. Fig. 9 shows the comparison of the zonal and meridional winds obtained by the Wuhan MST radar and the
285  radiosonde launched at 00 UT and 12 UT on 22 May 2016. In these figures and throughout the manuscript, the positive zonal
component corresponds to eastward wind, while the positive meridional component corresponds to northward wind. The
zonal wind profiles are in good agreement in the altitude ranges of 3.5 to 23 km. The meridional wind profiles also show
good agreement at most altitudes, but the measurements observed by the Wuhan MST radar are weaker around the height of
14 km. The underestimates of meridional winds could be due to the effect of aspect sensitivity (Thomas et al., 1997). The
small discrepancies at some heights may be attributed to the variations of atmospheric activities at different temporal and
spatial scales, and the different measurement principles and errors in both instruments are also significant reasons (Belu et al.,
2010; Hocking et al., 2001).

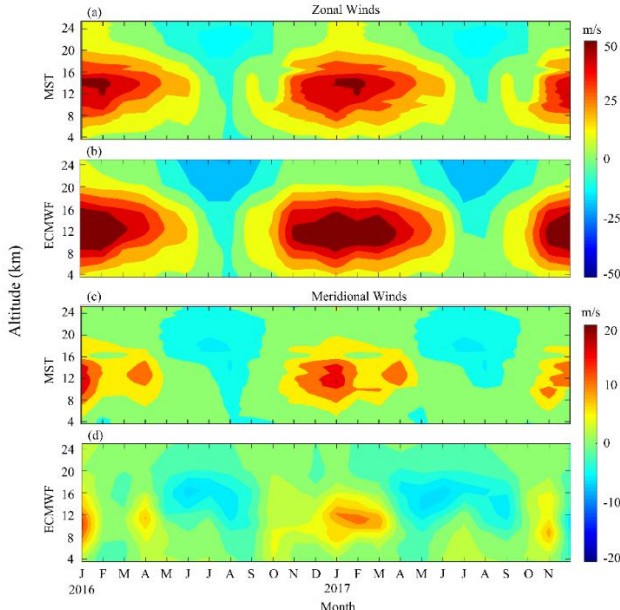

**Figure 10.** The contour plots of the monthly mean zonal (a) and meridional (c) winds in the troposphere and low stratosphere observed by
the Wuhan MST radar during Jan 2016-Dec 2017. ERA-interim model-estimated monthly mean zonal (b) and meridional (d) winds for the
Wuhan region during the same period

Fig. 10 shows the contour plots of the monthly mean zonal and meridional winds in the troposphere and low stratosphere
from the Wuhan MST radar and the ERA-interim. The observed mean winds are compared with the ERA-interim. ERA-
interim is the latest generation European Centre for Medium-Range Weather Forecasts (ECMWF) global atmospheric
reanalysis, which offers a good quality atmospheric wind with a 6 hour temporal resolution and 3° ×3°, 0.125° ×0.125°
latitude-longitude (Dee et al., 2011; Houchi et al., 2010). The dataset of monthly means of daily means is applied for the
present study, which is produced by the average of the four main synoptic monthly means at 00, 06, 12, and 18 UTC
(Berrisford et al., 2009).





It can be seen from Fig. 10(a) and Fig. 10(b) that the mean zonal wind observed by the Wuhan MST radar captures the major feature of the ERA-interim, which shows a clear annual oscillation with one westward jet and one eastward jet every year. The eastward jet occurs from September to June below ~20 km, and the westward jet occurs from May to October above ~20 km. The observed zonal winds in the eastward jet are ~10 m/s weaker than the ERA-interim reanalysis. The maximum magnitudes of the westward jet from the observation and the reanalysis are ~14 m/s and ~20 m/s, respectively. As

shown in Fig. 10(c) and Fig. 10(d), compared to the zonal winds, the meridional winds show larger discrepancies between the observation and the reanalysis. There are one northward jet and one southward jet exhibited in the observed mean meridional winds every year. The northward jet occurs from November to April below ~18 km, and the southward jet occurs from May to September above ~18 km. In the ERA-interim, the southward jets are extended in the two years. Especially in 2017, the southward jet occurs from April to October at almost the whole height, except in summer below ~12 km. The

discrepancies are mainly due to the differences of the average time periods. The meridional wind changes more over time, so as to show larger discrepancies in the monthly mean meridional winds. In conclusion, the Wuhan MST radar can measure the zonal and meridional winds in the troposphere and low stratosphere effectively.

### 3.3 Mesospheric observation

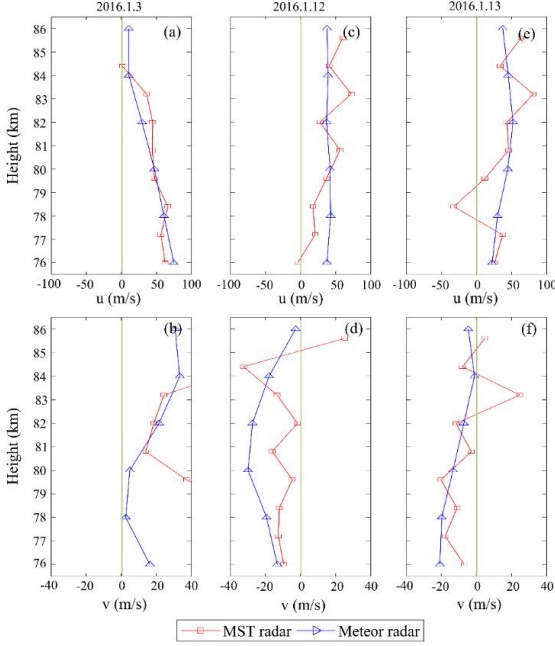

**Figure 11.** The daily mean zonal (u) and meridional (v) winds at the heights of 76 to 86 km observed by the Wuhan MST radar (red lines) and the Wuhan meteor radar (blue lines) on 3 January 2016 (a, b), 12 January 2016 (c, d) and 13 January 2016 (e, f).

In order to verify the validity of the mesospheric wind measurements, simultaneous observations obtained from the meteor radar at Wuhan were compared with the Wuhan MST radar observations. The Wuhan meteor radar (30.6°N, 114.4°E) is





325 about 120 km away from the Wuhan MST radar, which is an all-sky interferometric broadband radar system with a peak

power of 7.5 kW and a frequency of 38.7 MHz (Xiong et al., 2004; Zhao et al., 2005). The averaging of daytime (8 LT-16

LT) observations was used as the daily mean wind estimation for the Wuhan MST radar, while the 24 hours average was

used for the Wuhan meteor radar. Because of the effect of diurnal variations, the estimated mean winds of the Wuhan MST

may be biased less than 5 m/s compared with that of the Wuhan meteor radar below 85-90 km (Nakamura et al., 1996).

330 Considering the observation height range of the two radars, the comparison range was set at the heights of 76 to 86 km.

  Fig. 11 shows the daily mean zonal and meridional winds observed by the Wuhan MST radar and the Wuhan meteor radar

on 3 individual days in January 2016. Interestingly, the measurements at height of around 81 km show better agreement than

other heights. It is because the measurements of the meteor radar are more reliable above 80 km (Ratnam et al., 2001; Kumar

et al., 2008), while the data acquisition rate of the Wuhan MST radar is relatively high around the height of 80 km in the

335 mesosphere. From the comparison, the zonal and meridional winds are of concordance in the aggregate, whereas there are

some discrepancies. Two reasons might be resulting in the discrepancies between the observations of the two radars. The

first one is the localized gravity waves, tides or planetary waves could make the differences between them (Rao et al., 2014;

Ratnam et al., 2001). The second is that the low data acquisition rate of the Wuhan MST radar in the mesosphere could lead

to the fluctuations of the daily mean data, which shows the sudden changes of the MST radar measurements at some heights.

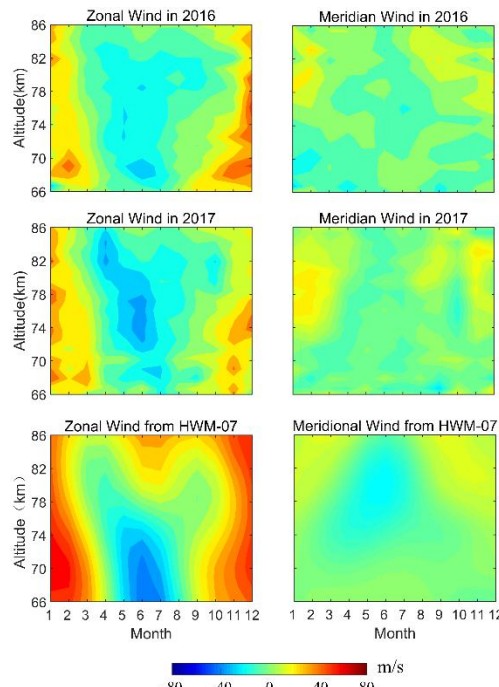


**Figure 12.** The contour plots of the monthly mean zonal (left) and meridional (right) winds in the mesosphere observed by the Wuhan
MST radar during Jan 2016-Dec 2017 and Jan 2017-Dec 2017. HWM-07 model-estimated winds (third row) for Wuhan region during the
same period.



Fig. 12 shows contour plots of the monthly mean zonal and meridional winds from the MST radar and HWM-07. The observed mean winds are compared with the Horizontal Wind Model-07 (HWM-07). HWM-07 synthesizes multiple instruments and considers various natural conditions, which can provide the atmosphere neutral winds from the ground to an altitude of 500 km (Drob et al., 2008).

As shown in left panels of Fig. 12, Wuhan MST radar zonal winds clearly show strong seasonal variations in 2016 and

2017. In general, the trends of the observational and predicted zonal winds match well, especially the reversal from eastward to westward in spring and the reversal from westward to eastward in autumn. However, HWM-07 overestimates the zonal winds largely in winter. The maximum magnitudes of the observational and predicted results are ~65 and ~51 m/s in winter. Many studies indicated that stronger northward and westward winds are happened after the stratospheric sudden warming (SSW) events (Mbatha et al., 2010; Chau et al., 2015), and this factor is not considered in the HWM-07. The SSW events

happened during the observation period may influence the mean winds in the mesosphere. Hence, the stronger westward winds may result in smaller mean zonal winds during winter. Moreover, the differences of the zonal winds in summer are noticed. The first difference is the reversal height in summer, which is a useful index for the mesopause. The wind shear around 78 km is prominent during the summer from the HWM-07. Meanwhile, the reversal height observed by the Wuhan MST radar is about 84-85 km, which is consistent with the result observed by the MU radar at similar altitude (Namboothiri

et al., 1999). Further study may be needed to analyze the difference. The second difference is the westward jet (the bluer region) occurred in summer. There are some differences of the westward jet between the observations and predictions in occurrence time and height, which could be due to interannual variability. As seen from right panels of Fig. 12, it appears that the observational meridional winds of 2016 and 2017 have the same trend as the predicted results. They all have one northward jet occurred above ~75 km in the period from August to April, but the observational results are larger than the

predicted results in winter because of the stronger northward winds during the SSW events. In general, the Wuhan MST radar wind measurements of the mesosphere are in agreement with the HWM-07 predictions in trend.

## 4 Conclusion

The technical features of the Wuhan MST radar are described in this manuscript. We use the TR modules and digital receiver with smart structure, and reasonable feeding network, to realize the beam steering for three-dimensional wind field

measurements. Short-term and long-term wind observations of the Wuhan MST radar are compared with other instruments and related models for validation, and the results are summarized as follows:

1. Compared with the radiosonde (120 km away) and the ERA-interim, the zonal and meridional winds are in good agreement at the heights of 3.5-25 km, and large discrepancies in meridional winds could be due to the temporal and spatial differences.

2. The daily mean zonal and meridional winds are in good agreement at the heights of 76 to 86 km with the Wuhan meteor radar (120 km away), and the measurements at height of around 81 km show better agreement than other heights.



3. The monthly mean zonal and meridional winds are in agreement with the HWM in trend at the heights of 66 to 86 km. The amplitudes of the observational results are different from that of the predicted results in winter, and it could be due to the influence of SSW events.

The comparisons indicate that the Wuhan MST is an effective tool to measure the three-dimensional wind fields of the MST region in the short-term and long-term. These results encourage us to do more for the improvements, such as improving the data acquisition of the high mode, and correcting the nominal zenith angle for the aspect sensitivity. In the future, we will use the Wuhan MST radar to study precipitation, gravity waves, and stratosphere-troposphere exchange processes during typhoon, cold front or other events, as well as the dynamics of the mesosphere.


*Data availability*. Wuhan MST radar data can be downloaded at http://159.226.22.74/.

*Author contributions*. LQ prepared the main part of the manuscript and performed the statistical analysis. GC is the project leader of the Wuhan MST radar and supported the preparation of the manuscript. SZ supervised the paper writing. QY
implemented the construction work. The measurements were led by WG and FC. WZ and HZ helped with the statistical analysis of Wuhan MST radar. MS and EL provided valuable suggestion for data processing. The data analysis was supported by XC and HS. HZ and LZ edited the article.

*Competing interests*. The authors declare that they have no conflict of interest.


*Acknowledgements*. The authors would like to acknowledge for the use of Wuhan MST radar data from the Chinese Meridian Project. This work is supported by the Natural Science Foundation of Zhejiang Province under Grant LQ20D040001, by the Chinese Meridian Project and National Natural Science Foundation of China under Grant 41722404.

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
