# Peer review of "Wuhan MST radar: Technical features and Validation of wind observations"

_Atmospheric Measurement Techniques, 2020_

## Short Comment (SC1) · 26 Feb 2020

General Comments:

This paper mainly describes the Wuhan MST radar system and verifies its observation results. I have read two published articles about the Wuhan MST radar, they are [Chen G. et al. 2016; Zhengyu, Z., et al. 2013]. I have tried to find some new information in this manuscript, but I really didn't find any new information that can meet the standards of an AMT publication. It is undeniable that the authors have done a lot of work, such as the system description looks very detailed, and used a variety of data to compare and verify with the observation results. However, this article is more like the instruction manual of the Wuhan MST. It contains a lot of content, but most of it is not clear and

there are many inaccuracies in this paper.

Specific Comments:

Fig.1-5: I think Chen et al. (2016) have made it clear, and it is also simple and easy to understand. The authors just repeated it in disguise, adding photos of some modules. Regarding this comment, I look forward to the authors' explanation.

Fig.6-7: Too foundational.

Line 266-269, Fig.8: Now that the authors indicate that the winds in the mesosphere are only available during the daytime, then why not separate the day- and night-time to get the data acquisition rate of the high mode. I strongly advised the authors to read more related literature about mesospheric echo.

Line 270-271: The maxmum data acquisition rate of only 10-17% (between âĹij68-82km region) is not enough to drawing conclusions that the Wuhan MST radar can effectively receive mesospheric echoes.

Fig.9: Why is the comparison result for only one case profile given? Only one profile comparison cannot even be expressed as short term comparison (Line 20). If the authors' intention is to verify the radar observations, a long-term comparison is necessary (maybe two years like Fig.10).

Fig.11: Now that the authors used the meteor radar observation data for comparison, that is to say, the authors recognizes the reliability of the meteor radar data, so why not make a longer time comparison ( like Figure 10 and Figure 12)? This is also necessary, both in terms of scientific rigor and the authors' own research purpose.

References:

Chen G., et al., MST Radars of Chinese Meridian Project: System Description and Atmospheric Wind Measurement. IEEE Transactions on Geoscience and Remote Sensing, 2016, 54(8): 4513-4523.

[Figure]

Zhengyu, Z., et al., Wuhan Atmosphere Radio Exploration (WARE) radar: System design and online winds measurements, Radio Sci., 2013, 48, 326–333, doi:10.1002/rds.20040.

---

## Referee Comment (RC1) · Anonymous Referee #1 · 27 Feb 2020

**Comments on the manuscript AMT-2020-17**

**General Comments:**

This paper introduces the technical features of the Wuhan MST radar, and shows the comparison of wind field with those measured by other instruments and related models for validation. This radar was initially established under the support of Meridian Project of China, and was upgraded in 2016. Another same type of MST radar supported by the Meridian Project of China is constructed in Xianghe, around Beijing of China. Two recent papers resulting from these MST radars are mentioned in this paper (2016: system description and wind measurement, and 2019: tropopause study). The present paper seems to introduce the upgraded radar system and validate the wind measurement by comparing with radiosonde, ERA-interim reanalysis, meteor wind, and HMW-07 model. A lot of work have been done.

Section 2 gives a detailed description of radar system and operation control. It is not easy to figure out the whole flowchart and circuits of design. I am not sure whether these detailed flowchart and circuits are suitable to published in AMT or not, although the design and construction of the radar system are worth being released for engineering reference.

According to the data collected in 2016 and 2017, the reliability of long-term wind field in the range interval of middle troposphere and lower stratosphere is higher than that in the mesosphere. The low data acquisition rate in the mesosphere seems to be the major problem, which also happens to other MST radars. In the last paragraph of section 3.1, it is mentioned that the winds in the mesosphere are only available during the daytime (8 LT-16 LT) in the D region (due to insufficient D region ionization during nighttime). I suggest that the authors also discuss the difference of turbulence scales in the lower and higher atmosphere, referring to Hocking (Radio Science, 20, p1410, 1985) or others.

In discussion of Fig. 12, the stratospheric sudden warming (SSW) event has been considered to be significant factor of some discrepancies between radar wind and HWM-07 model wind. Since the radar system works in low to high modes for 5 min in sequence, is it possible to examine the occurrence and prevailing rate of SSW events with the data of the low and middle modes or other information? In case the SSW events happen frequently during the observation period, it will provide explicit evidence of the discrepancies between radar wind and HWM-07 model wind. Could this evidence be included in this paper?

HWM-07 model was used in this study. However, there has been HWM-14 model. If possible, HWM-14 model can be employed instead of HWM-07.

**Other comments and suggestions:**

1) Fig. 1 is the schematic block of the radar system, and several paragraphs are written for this part. I suggest that the text for Fig. 1 can be section 2.1 (with suitable title). Section 2.1 becomes section 2.2, and so on.

2) L33-38: Many MST radars are mentioned here, and some of them have been upgraded, for example, the MU radar and Chung-li radar. Therefore, it is better to update some references.

3) L314: the southward jet occurs from April to October at almost the whole height, except in summer below ~12 km.
   Q: Do you mean "the southward jet occurs from April to October, and extends down to the low height in April and May." ?

4) L364-365: northward jet occurred above ~75 km in the period from August to April…during the SSW events.
   L378-379: …due to the influence of SSW events.
   Q: In fact, there is no evidence of SSW event shown in this paper to support the conclusion. Could this evidence be included in this paper?

5) L380: …is an effective tool to measure the three-dimensional wind fields…
   Q: The vertical wind is not shown in this paper. Do you also record the vertical wind velocity?

**Some wording problems are listed below for authors' reference, but please re-consider their suitability:**

1) L13: The radar system is …

2) L13: 192 kW or 172 kW?

3) L18, L40, L46, L59, L60,…: … paper.

4) L44: we plan to  write a new article…

5) L48: The location is far away from… (Do you mean this?)

6) L69: The shortest width of the subpulse …

7) L70: The radar system…

8) L88: …consists of the DDS (Direct Digital Synthesizer) module.

9) L96: wind  field…

10) L100: …wave  ratio (VSWM)…

11) L101: …Fig. 2 , there are …

12) L114: …, which respects the first…
    Q: Is the word "respects" proper here?

13) L117: By  analogy,…

14) L211: orthogonal phase (Q)…

I think the term "quadrature phase" is used commonly.

15) L221: … e.g. the 10 m…

16) L227: …Fig. 6(b), after …

17) L267: … 17%, which is much lower….middle modes.

18) L288: but the  winds observed …

19) L290: heights  could be attributed to…

20) L300: …generation of European…

21) L336: Two reasons might  result in the …

22) Fig. 12, caption: MST radar during Jan 2016-Dec 2016 and …

23) L353: …westward winds  happen after the…

24) …

---

## Referee Comment (RC2) · Anonymous Referee #2 · 12 Mar 2020

Review on Wuhan MST radar: Technical features and validation of wind observations, by Qiao

General comments: The main objectives of the manuscript are, i) describing the Wuhan MST radar system and ii) validating its measurements. The major problem of the manuscript is it neither describes the system completely nor does a comprehensive scientific evaluation of its products. It falls somewhere in between. Also, the radars built under Chinese Meridian project were discussed in earlier papers (see Chen et al. 2016). Then it is not clear to me what the authors want to describe/study here? The authors may focus on either complete description of the system (highlighting the updates from 2016 after Chen's publication, if any) or on a detailed scientific evaluation of products.

[Figure]

Specific comments: 1. The MST radars from China were discussed at length in few system related papers (Chen et al. 2016). What is new in this paper? Is there any upgrade made after those papers? If the authors intention is to highlight the stable performance of the system, then it is better to do a detailed scientific evaluation. 2. Lines 30-37: Several of these radars have been upgraded, like MU radar, Indian MST radar, NERC MST radar, etc. It is better to include recent references also to have updated knowledge on these radars. 3. The description of the system is not complete. Enough details were not provided on the antenna parameters, TR module specifications and RF performance. Also, it is better to include important specifications of the system in a table. 4. A separate sub-section exists on clutter suppression without describing how it is done! Is it simple removal of data at zero frequency and fill it with interpolated data from neighboring points? Or do you employ any filtering techniques (like wavelets)? 5. In spite of having two years of observations, the authors restricted the analysis to one profile comparison. Even that comparison shows a difference of 5-7 m s-1 in the mid- and upper-troposphere, too large to accept. The authors should do the validation using a large data set to have a statistically robust conclusion on the performance of the radar. 6. Line 289: Several reasons were quoted for the wind discrepancy, including aspect sensitivity, without dwelling on any of those issues. Mere quoting of some references (elsewhere) may not resolve the problems in your radar or analysis. If aspect sensitivity is the real reason, why is it occurring only at those heights and in meridional plane alone? 7. Line 308: Even the average wind difference between the radar and ERA is too large (10 ms-1). What could be the reason for this difference? Also, do some statistical analysis by providing RMSE and correlations with statistical significance tests. 8. Line 354: Same problem as above, the SSW events were cited as the potential reason for the wind discrepancy without verification. Instead of citing old references, why don't you check whether or not any such events occurred during that period? 9. So many grammatical errors to list here (few of them are given below in minor comments). They should be corrected before the submission of the revised version.

Minor Comments: Lines13-14: Rewrite these sentences. Line 26: Change to "The mesosphere-stratosphere-troposphere (MST) radars have been used for studying the......" Line 29: Replace 'applied' with 'employed' or some other suitable word. Same line, should be 'turbulence' Line 31: The sentence is abruptly ending. MST community plays a significant role in what? Lines 38-39: Rewrite these sentences. Line 44: Should be '….to write a new article in response to the readers and users demand (or request)...' Line 49: Remove 'of radar echoes' Line 74: The signal is scattered by 'refractive index irregularities'. Line 99: With 4 m antenna spacing, one can tilt the beam up to 24o from zenith without grating lobe!!. Line 114: 'respects' is not the correct word there. Line 115: ...data pots of .... Correct it. Line 154: How about azimuth angles? Line 174: The recovery time of T/R switch is somewhat on higher side, which restricts the minimum height coverage (if shorter pulses are available) Line 218: Replace 'in sunny day' with 'during fair weather' Line 210: Since the LNA bandwidth of small TR module is 1 MHz, FIR filter bandwidth of 1.5 MHz will not improve the performance. First of all, what is the logic in choosing 1 MHz bandwidth at LNA? Line 225: Should be 'Doppler spectra'. The sentences in this paragraph suffer with several grammatical errors. Correct them. Line 230: What do you mean by high-frequency interference? Line 231: Bring more clarity in presentation. At present, description of different modes of operation exists under 'Validation of wind observations'. Add one more subsection 3.1. Modes of operation and then change numbers of other subsections accordingly. Line 245: If the temporal resolution of the data is 30 min, then the number of data points in a day should be 48. Then how come different numbers for different modes? Line 280: The radiosonde generally take an hour to reach 18 km assuming an ascent rate of 5 m s-1. Is it a special sonde (or filled with more gas?) that reaches 25 km in 1 hour? Line 300: Which one is latest? ERA-Interim or ERA5? Line 334-336: Rewrite the sentences. Also the data acquisition rate is high at 75 km not at 80 km. Line 351-354: Rewrite the sentences.

---

## Author Comment (AC2) · 1 May 2020

General Comments:

Answer: Dear reviewer, this paper mainly focus on the system description of the Wuhan MST radar. Because Chen et al. (2016) has introduce the antenna array of the Wuhan MST radar, we mainly introduce the technical features in this paper. It includes antenna field, timing signal, TR module, digital transceiver and clutter suppression. Then we briefly analyze some cases and long tern comparisons. This part is the preliminary work of validation. We would like to thank the reviewer for valuable and constructive comments and suggestions. We have revised the paper in line with the reviewer's comments, thereby improving the technical quality and the clarity of the paper accordingly.

[Figure]

Specific Comments:

(1)The MST radars from China were discussed at length in few system related papers (Chen et al. 2016). What is new in this paper? Is there any upgrade made after those papers? If the authors intention is to highlight the stable performance of the system, then it is better to do a detailed scientific evaluation.

Answer: Dear reviewer, the RF circuits of TR modules were optimized, and the detailed description is shown in the paper. Meanwhile, the inter connections of the shelter and the feeding network were modified. Monitoring information of the small TR modules are shown in Figure 1. The red square represents the damaged TR module, and the green square represents the good TR module. It is obvious that the damaged TR modules decrease significantly after the upgrade. Therefore, the Wuhan MST radar is in good running condition after the upgrade.

(2)Lines 30-37: Several of these radars have been upgraded, like MU radar, Indian MST radar, NERC MST radar, etc. It is better to include recent references also to have updated knowledge on these radars.

Answer: Thank you for your suggestion. We updated the references in the revised paper.

(3)The description of the system is not complete. Enough details were not provided on the antenna parameters, TR module specifications and RF performance. Also, it is better to include important specifications of the system in a table.

Answer: Thank you for your suggestion. We listed the important specifications of the system in table 1 in the revised paper, including the antenna parameters, TR module specifications, RF performance and so on.

(4)A separate sub-section exists on clutter suppression without describing how it is done! Is it simple removal of data at zero frequency and fill it with interpolated data from neighboring points? Or do you employ any filtering techniques (like wavelets)?

Answer: Dear reviewer, the wavelet method is usually used in time-domain, and the ground clutter suppression used here is based on frequency-domain. The effectiveness of the algorithm is satisfactory.

(5)In spite of having two years of observations, the authors restricted the analysis to one profile comparison. Even that comparison shows a difference of 5-7 m s-1 in the mid- and upper-troposphere, too large to accept. The authors should do the validation using a large data set to have a statistically robust conclusion on the performance of the radar.

Answer: Dear reviewer, the radiosondes were launched by us on 22 May 2016, which are not from the standard observatory. Therefore, we don't have a large data set of the radiosondes, and it is difficult to do a long-term comparison between the Wuhan MST radar and the radiosonde. That's why we compare the mean zonal and meridional winds from the Wuhan MST radar and the ERA-interim, and the results are in good agreement at heights of 3.5-25 km. The difference of the results between the Wuhan MST radar and the radiosonde is due to the different measurement principles. The radiosonde will not just pass the detection area of the MST radar, and there is a difference of over 100 kilometers. So the comparison shows a difference of several meters, which is normal.

(6)Line 289: Several reasons were quoted for the wind discrepancy, including aspect sensitivity, without dwelling on any of those issues. Mere quoting of some references (elsewhere) may not resolve the problems in your radar or analysis. If aspect sensitivity is the real reason, why is it occurring only at those heights and in meridional plane alone?

Answer: Dear reviewer, the radiosondes were launched by us on 22 May 2016, which are not from the standard observatory. Therefore, there is only one profile comparison. Therefore, we can only introduce many possible reasons. The concrete reason needs more profile comparisons to analyze, which is our next work. I hope to get your

understanding.

(7)Line 308: Even the average wind difference between the radar and ERA is too large (10 ms-1). What could be the reason for this difference? Also, do some statistical analysis by providing RMSE and correlations with statistical significance tests.

Answer: The EAR radar is at the Indonesian equator (10.63°S), and the Wuhan MST radar is at latitude 29.5°N. The difference from the EAR radar observation is probably that the two radars are at different latitudes and in different atmospheric circulation. Considering the latitude difference, it is difficult to do the correlation analysis of the two radars. I hope to get your understanding.

(8)Line 354: Same problem as above, the SSW events were cited as the potential reason for the wind discrepancy without verification. Instead of citing old references, why don't you check whether or not any such events occurred during that period?

Answer: Thank you for your suggestion. Fig. 2(a) shows the time-altitude evolution of the daily mean zonal wind observed by the Wuhan MST radar from 66 to 86 km during 2016 SSW winter (Jan to Feb). The 2016 Feb SSW is a minor SSW, and the day of peak warming on Feb 5 is marked by the dotted vertical line. The wind weakening is observed around Feb 5. Note that the westward wind form 68 to 78 km during Jan 10 to Jan 14 is a reversal of the climatological mean zonal wind, which has nothing to do with the SSW. Fig. 2(b) shows the time-altitude evolution of the daily mean zonal wind observed by the Wuhan MST radar from 66 to 86 km during 2017 SSW winter (Jan to Feb). Two minor warming events happened during the winter of 2017, with two days of peak warming on Feb 2 and 26, marked by dotted vertical lines in the figure. The wind reversal is observed around Feb 2, and the wind weakening is observed around Feb 26 (not obvious). This is a preliminary analysis. Considering the discussion of SSW is not the gist of the paper, the figure will be used as a supplementary material.

(9) So many grammatical errors to list here (few of them are given below in minor comments). They should be corrected before the submission of the revised version.

[Figure]

Answer: Dear reviewer, we modified grammatical errors in the revised paper.

Minor Comments:

(1)Lines 13-14: Rewrite these sentences.

Answer: Thank you for your suggestion. We rewrite the sentences in the revised paper.

(2)Line 26: Change to "The mesosphere-stratosphere-troposphere (MST) radars have been used for studying the......"

Answer: Thank you for your suggestion. We modified it in the revised paper.

(3)Line 29: Replace 'applied' with 'employed' or some other suitable word. Same line, should be 'turbulence'

Answer: Thank you for your suggestion. We modified them in the revised paper.

(4)Line 31: The sentence is abruptly ending. MST community plays a significant role in what?

Answer: Thank you for your suggestion. We modified the sentence in the revised paper.

(5)Lines 38-39: Rewrite these sentences.

Answer: Thank you for your suggestion. We rewrite the sentences in the revised paper.

(6)Line 44: Should be '....to write a new article in response to the readers and users demand (or request)...'

Answer: Thank you for your suggestion. We modified the sentence in the revised paper.

(7)Line 49: Remove 'of radar echoes'

Answer: Thank you for your suggestion. We modified the sentence in the revised paper.
(8)Line 74: The signal is scattered by 'refractive index irregularities'.

Answer: Thank you for your suggestion. We modified the sentence in the revised paper.

(9)Line 99: With 4 m antenna spacing, one can tilt the beam up to 24° from zenith without grating lobe!!.

Answer: Thank you for your suggestion. We modified the angle in the revised paper.

(10)Line 114: 'respects' is not the correct word there.

Answer: Thank you for your suggestion. We modified the word in the revised paper.

(11)Line 115: ...data pots of .... Correct it.

Answer: Thank you for your suggestion. We modified it in the revised paper.

(12)Line 154: How about azimuth angles?

Answer: Dear reviewer, the oblique beams point to the north, south, west and east, and the azimuth angles corresponds to 90°, 180°, 270°, 360°.

(13)Line 174: The recovery time of T/R switch is somewhat on higher side, which restricts the minimum height coverage (if shorter pulses are available)

Answer: The minimum detecting height of the low mode is 3.5 km, and the propagation time (23 $\mu$s) is greater than the recovery time of T/R switch.

(14)Line 218: Replace 'in sunny day' with 'during fair weather'

Answer: Thank you for your suggestion. We modified it in the revised paper.

(15)Line 210: Since the LNA bandwidth of small TR module is 1 MHz, FIR filter bandwidth of 1.5 MHz will not improve the performance. First of all, what is the logic in choosing 1 MHz bandwidth at LNA?

Answer: Dear reviewer, the shortest pulse is 1 $\mu$s, so the LNA bandwidth is 1 MHz. We

made a mistake, and the FIR filter bandwidth is 1 MHz. We modified it in the revised paper.

(16)Line225: Should be 'Doppler spectra'. The sentences in this paragraph suffer with several grammatical errors. Correct them.

Answer: Thank you for your suggestion. We modified them in the revised paper.

(17)Line 230: What do you mean by high-frequency interference?

Answer: Dear reviewer, the high frequency interferences refer to internal noise of the radar.

(18) Line 231: Bring more clarity in presentation. At present, description of different modes of operation exists under 'Validation of wind observations'. Add one more subsection 3.1. Modes of operation and then change numbers of other subsections accordingly.

Answer: Thank you for your suggestion. We modified them in the revised paper.

(19)Line 245: If the temporal resolution of the data is 30 min, then the number of data points in a day should be 48. Then how come different numbers for different modes?

Answer: Dear reviewer, the Wuhan MST radar is down for maintenance periodically, and the radar system sometimes runs in the high mode. Therefore, there are different numbers for different modes.

(20)Line 280: The radiosonde generally take an hour to reach 18 km assuming an ascent rate of 5 m s-1. Is it a special sonde (or filled with more gas?) that reaches 25 km in 1 hour?

Answer: Dear reviewer, the radiosonde is the normal digital radiosonde filled with more gas.

(21)Line 300: Which one is latest? ERA-Interim or ERA5?

[Figure]

Answer: Dear reviewer, EAR-5 is latest, and we modified it in the revised paper.

(22)Line 334-336: Rewrite the sentences. Also the data acquisition rate is high at 75 km not at 80 km.

Answer: Thank you for your suggestion. We modified the sentences in the revised paper.

(23)Line 351-354: Rewrite the sentences.

Answer: Thank you for your suggestion. We modified the sentences in the revised paper.

Please also note the supplement to this comment:
https://www.atmos-meas-tech-discuss.net/amt-2020-17/amt-2020-17-AC2-supplement.pdf

[Figure]

Before the upgrade

Six months after the upgrade

[Figure]

(a)

(b)

**Fig. 1.** Monitoring information of the small TR modules.

Zonal Wind in 2016

(a)

Zonal Wind in 2017

(b)

**Fig. 2.** Time-altitude evolution of the daily mean zonal winds observed by the Wuhan MST radar from 66 to 86 km during Jan to Feb in 2016 (a) and 2017 (b). The dotted vertical lines indicate peak warming.

---

## Author Comment (AC3) · 1 May 2020

General Comments:

(1)I suggest that the authors also discuss the difference of turbulence scales in the lower and higher atmosphere, referring to Hocking (Radio Science, 20, p1410, 1985) or others.

Answer: Dear reviewer, radars with wide band are usually used to study the turbulence scales in the atmosphere. The Wuhan MST radar operates at fixed frequency, and it can provide only limited information about the turbulence scales. Therefore, we will not discuss the turbulence scales in the paper. I hope to get your understanding.

(2)In discussion of Fig. 12, the stratospheric sudden warming (SSW) event has been

considered to be significant factor of some discrepancies between radar wind and HWM-07 model wind. Since the radar system works in low to high modes for 5 min in sequence, is it possible to examine the occurrence and prevailing rate of SSW events with the data of the low and middle modes or other information?Could this evidence be included in this paper?

Answer: Thank you for your suggestion. Fig. 1(a) shows the time-altitude evolution of the daily mean zonal wind observed by the Wuhan MST radar from 66 to 86 km during 2016 SSW winter (Jan to Feb). The 2016 Feb SSW is a minor SSW, and the day of peak warming on Feb 5 is marked by the dotted vertical line. The wind weakening is observed around Feb 5. Note that the westward wind form 68 to 78 km during Jan 10 to Jan 14 is a reversal of the climatological mean zonal wind, which has nothing to do with the SSW. Fig. 1(b) shows the time-altitude evolution of the daily mean zonal wind observed by the Wuhan MST radar from 66 to 86 km during 2017 SSW winter (Jan to Feb). Two minor warming events happened during the winter of 2017, with two days of peak warming on Feb 2 and 26, marked by dotted vertical lines in the figure. The wind reversal is observed around Feb 2, and the wind weakening is observed around Feb 26 (not obvious). This is a preliminary analysis. Considering the discussion of SSW is not the gist of the paper, the figure will be used as a supplementary material.

(3)HWM-14 module can be employed instead of HWM-07.

Answer: Thank you for your suggestion. We have employed HWM-14 instead of HWM-07. ActuallyïijŇthe predicted winds from HWM-14 is closer to the observations.

Other comments and suggestions:

(1)Fig. 1 is the schematic block of the radar system, and several paragraphs are written for this part. I suggest that the text for Fig. 1 can be section 2.1 (with suitable title). Section 2.1 becomes section 2.2, and so on.

Answer: Thank you for your suggestion. We modified the section titles in the revised

paper.

(2)L33-38: Many MST radars are mentioned here, and some of them have been up-graded, for example, the MU radar and Chung-li radar. Therefore, it is better to update some references.

Answer: Thank you for your suggestion. We updated the references in the revised paper.

(3)L314: the southward jet occurs from April to October at almost the whole height, except in summer below ∼12 km. Q: Do you mean"the southward jet occurs from April to October, and extends down to the low height in April and May." ?

Answer: Yes, that is what we mean. We modified the sentence in the revised paper.

(4)L364-365: northward jet occurred above ∼75 km in the period from August to April. . .during the SSW events. L378-379: . . .due to the influence of SSW events. Q: In fact, there is no evidence of SSW event shown in this paper to support the conclusion. Could this evidence be included in this paper?

Answer: Dear reviewer, the evidence of SSW event is shown in figure 1 of the response, and related references have been listed in the revised paper.

(5)L380: . . .is an effective tool to measure the three-dimensional wind fields. . . Q: The vertical wind is not shown in this paper. Do you also record the vertical wind velocity?

Answer: Dear reviewer, the main objective of the paper is validating its measurements. The vertical wind velocity is very small, and there is no appropriate data to verify its effectiveness. Therefore, we didn't discuss the vertical wind in this paper.

Some wording problems:

(1)L13: The radar system is . . .

Answer: Thank you for your suggestion. We modified it in the revised paper.

[Figure]

(2)192 kW or 172 kW?

Answer: Thank you for your suggestion. We modified it in the revised paper.

(3)L18, L40, L46, L59, L60,. . .: . . .paper paper.

Answer: Thank you for your suggestion. We modified them in the revised paper.

(4)L44: we plan to wright write a new article. . .

Answer: Thank you for your suggestion. We modified it in the revised paper.

(5)L48: The location is far away from. . . (Do you mean this?)

Answer: Thank you for your suggestion. We modified it in the revised paper.

(6)L69: The shortest width of the subpulse width. . .

Answer: Thank you for your suggestion. We modified it in the revised paper.

(7)L70: The radar system. . .

Answer: Thank you for your suggestion. We modified it in the revised paper.

(8)L88: . . .consists of the DDS (Direct Digital Synthesizer) module.

Answer: We have explained the DDS in L85.

(9)L96: wind filed field. . .

Answer: Thank you for your suggestion. We modified it in the revised paper.

(10)L100: . . .wave radio ratio (VSWM). . .

Answer: Thank you for your suggestion. We modified it in the revised paper.

(11)L101: . . .Fig. 2 for e.g. S0101, there are . . .

Answer: Thank you for your suggestion. We modified it in the revised paper.

(12)L114: . . ., which respects the first. . . Q: Is the word "respects" proper here?

Answer: Thank you for your suggestion. We modified it in the revised paper.

(13)L117: By that analogy,. . .

Answer: Thank you for your suggestion. We modified it in the revised paper.

(14)L211: orthogonal phase (Q). . .I think the term "quadrature phase" is used commonly.

Answer: Thank you for your suggestion. We modified it in the revised paper.

(15)L221: . . .i.e. e.g. the 10 m. . .

Answer: Thank you for your suggestion. We modified it in the revised paper.

(16)L227: . . .Fig. 6(b), Aafter . . .

Answer: Thank you for your suggestion. We modified it in the revised paper.

(17)L267: . . . 17%, which is much lower. . ..middle modes.

Answer: Thank you for your suggestion. We modified it in the revised paper.

(18)L288: but the measurement winds observed . . .

Answer: Thank you for your suggestion. We modified it in the revised paper.

(19)L290: heights may could be attributed to. . .

Answer: Thank you for your suggestion. We modified it in the revised paper.

(20)L300: . . .generation of European. . .

Answer: Thank you for your suggestion. We modified it in the revised paper.

(21)L336: Two reasons might be resulting in the . . .

Answer: Thank you for your suggestion. We modified it in the revised paper.

(22)Fig. 12, caption: MST radar during Jan 2016-Dec 20176 and . . .

Answer: Thank you for your suggestion. We modified it in the revised paper.

(23)L353: . . .westward winds are happened after the. . .

Answer: Thank you for your suggestion. We modified it in the revised paper.

Please also note the supplement to this comment:
https://www.atmos-meas-tech-discuss.net/amt-2020-17/amt-2020-17-AC3-supplement.pdf
* * *
[Figure]

Zonal Wind in 2016

(a)

Zonal Wind in 2017

(b)

Altitude(km)

Day

m/s

**Fig. 1.** Time-altitude evolution of the daily mean zonal winds observed by the Wuhan MST radar from 66 to 86 km during Jan to Feb in 2016 (a) and 2017 (b). The dotted vertical lines indicate peak warming.

**Supplement:**

[revised manuscript text omitted]

---

## Author Response (AR1)

**Response to review comments of amt-2020-17**

Wuhan MST radar: Technical features and Validation of wind observations

Lei Qiao, Gang Chen, Shaodong Zhang, Qi Yao, Wanlin Gong, Mingkun Su, Feilong Chen, Erxiao Liu, Weifan Zhang, Huangyuan Zeng, Xuesi Cai, Huina Song, Huan Zhang, Liangliang Zhang

May 22, 2020

Dear Editor:

Please find enclosed the revision of our submission "Wuhan MST radar: Technical features and Validation of wind observations" (ID: amt-2020-17).

We would like to thank you for handling the review process of our paper. We are also indebted to the reviewers for their helpful comments. In this revision, all of the comments raised have been addressed and marked in the revised manuscript. A detailed point-by-point response to the comments is given below.

We appreciate for Editors/Reviewers' warm work earnestly, and hope that the correction will meet with approval. Once again, thank you very much for your comments and suggestions.

Yours sincerely,
Lei Qiao

**Note:** To help legibility of the remainder of this response letter, all the reviewers' comments and questions are written in black color. Our responses and remarks are written in blue color.

**Response to RC1**

**General Comments:**

**Question (1)**

I suggest that the authors also discuss the difference of turbulence scales in the lower and higher atmosphere, referring to Hocking (Radio Science, 20, p1410, 1985) or others.

**Answer:** Dear reviewer, radars with wide band are usually used to study the turbulence scales in the atmosphere. The Wuhan MST radar operates at fixed frequency, and it can provide only limited information about the turbulence scales. Therefore, we will not discuss the turbulence scales in the paper. I hope to get your understanding.

**Question (2)**

In discussion of Fig. 12, the stratospheric sudden warming (SSW) event has been considered to be significant factor of some discrepancies between radar wind and HWM-07 model wind. Since the radar system works in low to high modes for 5 min in sequence, is it possible to examine the occurrence and prevailing rate of SSW events with the data of the low and middle modes or other information? Could this evidence be included in this paper?

**Answer:** Thank you for your suggestion. Fig. 1(a) shows the time-altitude evolution of the daily mean zonal wind observed by the Wuhan MST radar from 66 to 86 km during 2016 SSW winter (Jan to Feb). The 2016 Feb SSW is a minor SSW, and the day of peak warming on Feb 5 is marked by the dotted vertical line. The wind weakening is observed around Feb 5. Note that the westward wind form 68 to 78 km during Jan 10 to Jan 14 is a reversal of the climatological mean zonal wind, which has nothing to do with the SSW. Fig. 1(b) shows the time-altitude evolution of the daily mean zonal wind observed by the Wuhan MST radar from 66 to 86 km during 2017 SSW winter (Jan to Feb). Two minor warming events happened during the winter of 2017, with two days of peak warming on Feb 2 and 26, marked by dotted vertical lines in the figure. The wind reversal is observed around Feb 2, and the wind weakening is observed around Feb 26 (not obvious). This is a preliminary analysis. Considering the discussion of SSW is not the gist of the paper, the figure will be used as a supplementary material. We added related explanation in Lines 369-371 in the

revised paper.

[Figure]

**Figure 1** Time-altitude evolution of the daily mean zonal winds observed by the Wuhan MST radar from 66 to 86 km during Jan to Feb in 2016 (a) and 2017 (b). The dotted vertical lines indicate the days of peak warming.

**Question (3)**

HWM-14 module can be employed instead of HWM-07.

**Answer:** Thank you for your suggestion. We have employed HWM-14 instead of HWM-07. Actually,the predicted winds from HWM-14 is closer to the observations. See lines 360-383.

**Other comments and suggestions:**

**Question (1)**

Fig. 1 is the schematic block of the radar system, and several paragraphs are written for this part. I suggest that the text for Fig. 1 can be section 2.1 (with suitable title). Section 2.1 becomes section 2.2, and so on.

**Answer:** Thank you for your suggestion. We modified the section titles in the revised

paper.

**Question (2)**

L33-38: Many MST radars are mentioned here, and some of them have been upgraded, for example, the MU radar and Chung-li radar. Therefore, it is better to update some references.

**Answer:** Thank you for your suggestion. We updated the references in the revised paper. See lines 34-36.

**Question (3)**

L314: the southward jet occurs from April to October at almost the whole height, except in summer below ~12 km.

Q: Do you mean"the southward jet occurs from April to October, and extends down to the low height in April and May." ?

**Answer:** Yes, that is what we mean. We modified the sentence in the revised paper. See line 329.

**Question (4)**

L364-365: northward jet occurred above ~75 km in the period from August to April…during the SSW events.

L378-379: …due to the influence of SSW events.

Q: In fact, there is no evidence of SSW event shown in this paper to support the conclusion. Could this evidence be included in this paper?

**Answer:** Dear reviewer, the evidence of SSW event is shown in figure 1 of the response, and related analysis have been made in the revised paper. See lines 369-371.

**Question (5)**

L380: …is an effective tool to measure the three-dimensional wind fields…

Q: The vertical wind is not shown in this paper. Do you also record the vertical wind velocity?

**Answer:** Dear reviewer, the main objective of the paper is validating its measurements. The vertical wind velocity is very small, and there is no appropriate data to verify its effectiveness. Therefore, we didn't discuss the vertical wind in this paper. I hope to get your understanding.

**Other comments and suggestions:**

(1) L13: The radar system is …

**Answer:** Thank you for your suggestion. We modified it in the revised paper. See line 13.

(2) 192 kW or 172 kW?

**Answer:** Thank you for your suggestion. We modified it in the revised paper. See line 13.

(3) L18, L40, L46, L59, L60,…: … paper.

**Answer:** Thank you for your suggestion. We modified them in the revised paper. See lines 18, 40, 46, 59, 61, 300, 385, 405, and 406.

(4) L44: we plan to  write a new article…

**Answer:** Thank you for your suggestion. We modified it in the revised paper. See line 44.

(5) L48: The location is far away from… (Do you mean this?)

**Answer:** Thank you for your suggestion. We modified it in the revised paper. See line 48.

(6) L69: The shortest width of the subpulse …

**Answer:** Thank you for your suggestion. We modified it in the revised paper. See line 69.

(7) L70: The radar system…

**Answer:** Thank you for your suggestion. We modified it in the revised paper. See line 70.

(8) L88: …consists of the DDS (Direct Digital Synthesizer) module.

**Answer:** We have explained the DDS in line 90.

(9) L96: wind  field…

**Answer:** Thank you for your suggestion. We modified it in the revised paper. See line 101.

(10) L100: …wave  ratio (VSWM)…

**Answer:** Thank you for your suggestion. We modified it in the revised paper. See line 108.

(11)  L101: …Fig. 2 , there are …

**Answer:** Thank you for your suggestion. We modified it in the revised paper. See line 109.

(12)  L114: …, which respects the first… Q: Is the word "respects" proper here?

**Answer:** Thank you for your suggestion. We modified it in the revised paper. See line 118.

(13)  L117: By  analogy,…

**Answer:** Thank you for your suggestion. We modified it in the revised paper. See line 121.

(14) L211: orthogonal phase (Q)…I think the term "quadrature phase" is used commonly.

**Answer:** Thank you for your suggestion. We modified it in the revised paper. See line 216.

(15)  L221: … e.g. the 10 m…

**Answer:** Thank you for your suggestion. We modified it in the revised paper. See line 227.

(16)  L227: …Fig. 6(b), after …

**Answer:** Thank you for your suggestion. We modified it in the revised paper. See line 233.

(17)  L267: … 17%, which is much lower….middle modes.

**Answer:** Thank you for your suggestion. We modified it in the revised paper. See line 282.

(18)  L288: but the  winds observed …

**Answer:** Thank you for your suggestion. We modified it in the revised paper. See line 303.

(19)  L290: heights  could be attributed to…

**Answer:** Thank you for your suggestion. We modified it in the revised paper. See line 305.

(20) L300: …generation of European…

**Answer:** Thank you for your suggestion. We modified it in the revised paper. See line 315.

(21) L336: Two reasons might be resulting in the …

**Answer:** Thank you for your suggestion. We modified it in the revised paper. See line 350.

(22) Fig. 12, caption: MST radar during Jan 2016-Dec 20176 and …

**Answer:** Thank you for your suggestion. We modified it in the revised paper. See line 357.

(23) L353: …westward winds are happened after the…

**Answer:** Thank you for your suggestion. We modified it in the revised paper. See line 368.

**Response to RC2**

**General Comments:**

**Answer:** Dear reviewer, this paper mainly focus on the system description of the Wuhan MST radar. Because Chen et al. (2016) has introduced the antenna array of the Wuhan MST radar, we mainly introduce the technical features in this paper. It includes antenna field, timing signal, TR module, digital transceiver and clutter suppression. Then we briefly analyze some cases and long term comparisons. This part is the preliminary work of validation.

We would like to thank the reviewer for valuable and constructive comments and suggestions. We have revised the paper in line with the reviewer's comments, thereby improving the technical quality and the clarity of the paper accordingly.

**Specific Comments:**

**Question (1)**

The MST radars from China were discussed at length in few system related papers (Chen et al. 2016). What is new in this paper? Is there any upgrade made after those papers? If the authors intention is to highlight the stable performance of the system, then it is better to do a detailed scientific evaluation.

**Answer:** Dear reviewer, the RF circuits of TR modules were optimized, and the detailed description is shown in the paper. Meanwhile, the inter connections of the shelter and the feeding network were modified. Monitoring information of the small TR modules are shown in Figure 1 in the response. The red square represents the damaged TR module, and the green square represents the good TR module. It is obvious that the damaged TR modules decrease significantly after the upgrade. Therefore, the Wuhan MST radar is in good running condition after the upgrade, and the overall operation conditions are shown in Figure 7 in the paper.

[Figure]

Figure 1. Monitoring information of the small TR modules

**Question (2)**

Lines 30-37: Several of these radars have been upgraded, like MU radar, Indian MST radar, NERC MST radar, etc. It is better to include recent references also to have updated knowledge on these radars.

**Answer:** Thank you for your suggestion. We updated the references in the revised paper. See lines 34-36.

**Question (3)**

The description of the system is not complete. Enough details were not provided on the antenna parameters, TR module specifications and RF performance. Also, it is better to include important specifications of the system in a table.

**Answer:** Thank you for your suggestion. We listed the important specifications of the system in table 1 in the revised paper, including the antenna parameters, TR module specifications, RF performance and so on. See line 75.

**Question (4)**

A separate sub-section exists on clutter suppression without describing how it is done! Is it simple removal of data at zero frequency and fill it with interpolated data from neighboring points? Or do you employ any filtering techniques (like wavelets)?

**Answer:** Dear reviewer, the wavelet method is usually used in time-domain, and the ground clutter suppression used here is based on frequency-domain. The effectiveness of the algorithm is satisfactory, and the results are shown in Figure 6.

**Question (5)**

In spite of having two years of observations, the authors restricted the analysis to one

profile comparison. Even that comparison shows a difference of 5-7 m s-1 in the mid- and upper-troposphere, too large to accept. The authors should do the validation using a large data set to have a statistically robust conclusion on the performance of the radar.

**Answer:** Dear reviewer, the radiosondes were launched by us on 22 May 2016, which are not from the standard observatory. Therefore, we don't have a large data set of the radiosondes, and it is difficult to do a long-term comparison between the Wuhan MST radar and the radiosonde. That's why we compare the mean zonal and meridional winds from the Wuhan MST radar and the ERA-interim, and the results are in good agreement at heights of 3.5-25 km. The difference of the results between the Wuhan MST radar and the radiosonde is due to the different measurement principles. The radiosonde will not just pass the detection area of the MST radar, and there is a difference of over 100 kilometers. Meanwhile, the meridional winds change more than the zonal winds. So the comparison shows a difference of several meters, which is normal.

**Question (6)**

Line 289: Several reasons were quoted for the wind discrepancy, including aspect sensitivity, without dwelling on any of those issues. Mere quoting of some references (elsewhere) may not resolve the problems in your radar or analysis. If aspect sensitivity is the real reason, why is it occurring only at those heights and in meridional plane alone?

**Answer:** Dear reviewer, the radiosondes were launched by us on 22 May 2016, which are not from the standard observatory. Therefore, there is only one profile comparison. Therefore, we can only introduce many possible reasons. The concrete reason needs more profile comparisons to analyze, which is our next work. I hope to get your understanding.

**Question (7)**

Line 308: Even the average wind difference between the radar and ERA is too large (10 ms-1). What could be the reason for this difference? Also, do some statistical analysis by providing RMSE and correlations with statistical significance tests.

**Answer:** The EAR radar is at the Indonesian equator (10.63°S), and the Wuhan MST radar is at latitude 29.5°N. The difference from the EAR radar observation is probably that the two radars are at different latitudes and in different atmospheric circulation. Considering the latitude difference, it is difficult to do the correlation analysis of the two radars. I hope to get your understanding.

**Question (8)**

Line 354: Same problem as above, the SSW events were cited as the potential reason for the wind discrepancy without verification. Instead of citing old references, why don't you check whether or not any such events occurred during that period?

Answer: Thank you for your suggestion. Fig. 2(a) shows the time-altitude evolution of the daily mean zonal wind observed by the Wuhan MST radar from 66 to 86 km during 2016 SSW winter (Jan to Feb). The 2016 Feb SSW is a minor SSW, and the day of peak warming on Feb 5 is marked by the dotted vertical line. The wind weakening is observed around Feb 5. Note that the westward wind form 68 to 78 km during Jan 10 to Jan 14 is a reversal of the climatological mean zonal wind, which has nothing to do with the SSW. Fig. 2(b) shows the time-altitude evolution of the daily mean zonal wind observed by the Wuhan MST radar from 66 to 86 km during 2017 SSW winter (Jan to Feb). Two minor warming events happened during the winter of 2017, with two days of peak warming on Feb 2 and 26, marked by dotted vertical lines in the figure. The wind reversal is observed around Feb 2, and the wind weakening is observed around Feb 26 (not obvious). This is a preliminary analysis. Considering the discussion of SSW is not the gist of the paper, the figure will be used as a supplementary material. We added related explanation in Lines 369-371 in the revised paper.

[Figure]

**Figure 2** Time-altitude evolution of the daily mean zonal winds observed by the Wuhan MST radar from 66 to 86 km during Jan to Feb in 2016 (a) and 2017 (b). The dotted vertical lines indicate the days of peak warming.

**Question (9)**

So many grammatical errors to list here (few of them are given below in minor comments). They should be corrected before the submission of the revised version.

**Answer:** Dear reviewer, we have modified the grammatical errors in the revised paper.

**Minor Comments:**

(1) Lines 13-14: Rewrite these sentences.

**Answer:** Thank you for your suggestion. We rewrite the sentences in the revised paper. See lines 13-14.

(2) Line 26: Change to "The mesosphere-stratosphere-troposphere (MST) radars have been used for studying the......"

**Answer:** Thank you for your suggestion. We modified it in the revised paper. See line 25.

(3) Line 29: Replace 'applied' with 'employed' or some other suitable word. Same line, should be 'turbulence'

**Answer:** Thank you for your suggestion. We modified them in the revised paper. See line 28.

(4) Line 31: The sentence is abruptly ending. MST community plays a significant role in what?

**Answer:** Thank you for your suggestion. We modified the sentence in the revised paper. See line 30.

(5) Lines 38-39: Rewrite these sentences.

**Answer:** Thank you for your suggestion. We rewrite the sentences in the revised paper. See lines 38-39.

(6) Line 44: Should be '....to write a new article in response to the readers and users demand (or request)...'

**Answer:** Thank you for your suggestion. We modified the sentence in the revised paper. See lines 44-45.

(7) Line 49: Remove 'of radar echoes'

**Answer:** Thank you for your suggestion. We modified the sentence in the revised paper. See line 49.

(8) Line 74: The signal is scattered by 'refractive index irregularities'.

**Answer:** Thank you for your suggestion. We modified the sentence in the revised paper. See line 79.

(9) Line 99: With 4 m antenna spacing, one can tilt the beam up to 24° from zenith without grating lobe!!.

**Answer:** Thank you for your suggestion. We modified the angle in the revised paper. See line 107.

(10) Line 114: 'respects' is not the correct word there.

**Answer:** Thank you for your suggestion. We modified the word in the revised paper. See line 118.

(11) Line 115: ...data pots of .... Correct it.

**Answer:** Thank you for your suggestion. We modified it in the revised paper. See line 119.

(12) Line 154: How about azimuth angles?

**Answer:** Dear reviewer, the oblique beams point to the north, south, west and east, and the azimuth angles corresponds to 90°, 180°, 270°, 360°.

(13) Line 174: The recovery time of T/R switch is somewhat on higher side, which restricts the minimum height coverage (if shorter pulses are available)

**Answer:** The minimum detecting height of the low mode is 3.5 km, and the propagation time (23 µs) is greater than the recovery time of T/R switch.

(14) Line 218: Replace 'in sunny day' with 'during fair weather'

**Answer:** Thank you for your suggestion. We modified it in the revised paper. See line 224.

(15) Line 210: Since the LNA bandwidth of small TR module is 1 MHz, FIR filter bandwidth of 1.5 MHz will not improve the performance. First of all, what is the logic in choosing 1 MHz bandwidth at LNA?

**Answer:** Dear reviewer, the shortest pulse is 1 µs, so the LNA bandwidth is 1 MHz.

We made a mistake, and the FIR filter bandwidth is 1 MHz. We modified it in the revised paper. See lines 210 and 215.

(16) Line225: Should be 'Doppler spectra'. The sentences in this paragraph suffer with several grammatical errors. Correct them.

**Answer:** Thank you for your suggestion. We modified them in the revised paper. See lines 231, 232 and 236.

(17) Line 230: What do you mean by high-frequency interference?

**Answer:** Dear reviewer, the high frequency interferences refer to internal noise of the radar.

(18) Line 231: Bring more clarity in presentation. At present, description of different modes of operation exists under 'Validation of wind observations'. Add one more subsection 3.1. Modes of operation and then change numbers of other subsections accordingly.

**Answer:** Thank you for your suggestion. We modified them in the revised paper. See line 240.

(19) Line 245: If the temporal resolution of the data is 30 min, then the number of data points in a day should be 48. Then how come different numbers for different modes?

**Answer:** Dear reviewer, the Wuhan MST radar is down for maintenance periodically, and the radar system sometimes runs only in the high mode. Therefore, there are different numbers for different modes.

(20) Line 280: The radiosonde generally take an hour to reach 18 km assuming an ascent rate of 5 m s-1. Is it a special sonde (or filled with more gas?) that reaches 25 km in 1 hour?

**Answer:** Dear reviewer, the radiosonde is the normal digital radiosonde filled with more gas.

(21) Line 300: Which one is latest? ERA-Interim or ERA5?

**Answer:** Dear reviewer, EAR-5 is latest, and we modified it in the revised paper. See line 315.

(22) Line 334-336: Rewrite the sentences. Also the data acquisition rate is high at 75

km not at 80 km.

**Answer:** Thank you for your suggestion. We modified the sentences in the revised paper. See lines 349-351.

(23) Line 351-354: Rewrite the sentences.

**Answer:** Thank you for your suggestion. We modified the sentences in the revised paper. See lines 366-369

**Response to SC1**

**General Comments:**

**Answer:** Dear reviewer, this paper mainly focus on the system description of the Wuhan MST radar. Because Chen et al. (2016) has introduce the antenna array of the Wuhan MST radar, we mainly introduce the technical features in this paper. The RF circuits of the TR modules and the feeding network were optimized in the upgrade. Then we analyze the mesospheric echoes in section 3.3, and this is the first time to show the mesospheric observation of the Wuhan MST radar.

We would like to thank the reviewer for valuable and constructive comments and suggestions. We have revised the paper in line with the reviewer's comments, thereby improving the technical quality and the clarity of the paper accordingly.

**Specific Comments:**

**Question (1)**

Fig.1-5: I think Chen et al. (2016) have made it clear, and it is also simple and easy to understand. The authors just repeated it in disguise, adding photos of some modules. Regarding this comment, I look forward to the authors' explanation.

**Answer:** Dear reviewer, Chen et al. (2016) briefly introduced the antenna array of the Beijing MST radar. In this paper, the RF circuits of the small TR modules and big TR modules are optimized, and the detailed description is shown in the paper. The inter connections of the shelter and the feeding network are modified. Meanwhile, this paper introduces related timing signals and digital transceiver. We modified some sentences in the introduction.

**Question (2)**

Fig.6-7: Too foundational.

**Answer:** Dear reviewer, figure 6 shows the processing procedure. Although it is very basic, it is the important part of signal processing. Figure 7 shows the monthly total number of the Wuhan MST radar data in three observation modes, which shows good running condition of the Wuhan MST radar. Therefore, the two figures are necessary.

**Question (3)**

Line 266-269, Fig.8: Now that the authors indicate that the winds in the mesosphere

are only available during the daytime, then why not separate the day- and night-time to get the data acquisition rate of the high mode. I strongly advised the authors to read more related literature about mesospheric echo.

**Answer:** Thank you for your suggestion. We have read some related literature about mesospheric echo. The low data acquisition rate in the mesosphere also happens to other MST radars. We have separated the day- and night-time data acquisition rate of the high mode. Unfortunately, there is hardly any mesospheric echo during the nighttime. Therefore, we show the average data acquisition rate of the high mode throughout the day.

**Question (4)**

Line 270-271: The maximum data acquisition rate of only 10-17% (between 68-82km region) is not enough to drawing conclusions that the Wuhan MST radar can effectively receive mesospheric echoes.

**Answer:** Dear reviewer, we may not explain clearly. The data acquisition rate of 10-17% is on average throughout the day, and the maximum data acquisition during the daytime is more than 50%. Meanwhile, the daily mean zonal and meridional winds are in good agreement at the heights of 76 to 86 km with the Wuhan meteor radar. The monthly mean zonal and meridional winds are in agreement with the HWM in trend at the heights of 66 to 86 km. Therefore, it can prove that the Wuhan MST radar can effectively receive mesospheric echoes.

**Question (5)**

Fig.9: Why is the comparison result for only one case profile given? Only one profile comparison cannot even be expressed as short term comparison (Line 20). If the authors' intention is to verify the radar observations, a long-term comparison is necessary (maybe two years like Fig.10).

**Answer:** Dear reviewer, we may not express accurately. One profile comparison can't really be expressed as short term comparison. However, the radiosondes were launched by us on 22 May 2016, which are not from the standard observatory. Therefore, we don't have a large data set of the radiosondes, and it is difficult to do a long-term comparison between the Wuhan MST radar and the radiosonde. That's why we compare the mean zonal and meridional winds from the Wuhan MST radar and the ERA-interim, and the results are in good agreement at heights of 3.5-25 km. Therefore, the comparison is just one case, and the case can also indicate the Wuhan MST radar is an effective tool to measure wind fields. We modified the sentence in the revised paper.

**Question (6)**

Fig.11: Now that the authors used the meteor radar observation data for comparison, that is to say, the authors recognizes the reliability of the meteor radar data, so why not make a longer time comparison ( like Figure 10 and Figure 12)? This is also necessary, both in terms of scientific rigor and the authors' own research purpose.

**Answer:** Thank you for your suggestion. We also want to make a longer time comparison, but it is hard to realize. We made the simultaneous observation from 3 January 2016 to 13 January 2016, and there are only 3 days valid data. Therefore, we can only do case study, which is common for the comparison between the MST radar and the meteor radar (Rao et al., 2014). The three cases also indicate that 
[revised manuscript text omitted]

---

## Referee Report (RR1)

[referee-annotated manuscript omitted]

---

## Author Response (AR2)

**Response to review comments of amt-2020-17**

Wuhan MST radar: Technical features and Validation of wind observations

Lei Qiao, Gang Chen, Shaodong Zhang, Qi Yao, Wanlin Gong, Mingkun Su,
Feilong Chen, Erxiao Liu, Weifan Zhang, Huangyuan Zeng, Xuesi Cai, Huina Song,
Huan Zhang, Liangliang Zhang

July 17, 2020

Dear Editor:

Please find enclosed the revision of our submission "Wuhan MST radar: Technical features and Validation of wind observations" (ID: amt-2020-17).

We would like to thank you for handling the review process of our paper. We are also indebted to the reviewers for their helpful comments. In this revision, all of the comments raised have been addressed and marked in the revised manuscript. A detailed point-by-point response to the comments is given below.

We appreciate for Editors/Reviewers' warm work earnestly, and hope that the correction will meet with approval. Once again, thank you very much for your comments and suggestions.

Yours sincerely,
Lei Qiao

**Note:** To help legibility of the remainder of this response letter, all the reviewers' comments and questions are written in black color. Our responses and remarks are written in blue color.

**Response to Referee #1**

**Question (1)**

I suggest that the authors discuss the difference of turbulence scales in the lower and higher atmosphere, referring to Hocking (Radio Science, 20, p1410, 1985) or others. It is not a demand for the authors to use wide-band radars to study the turbulence scales. As indicated in Hocking's paper, the internal scales of neutral air turbulence are between several meters to tens of meters in the mesosphere, which are mostly larger than a half of the radio wavelength at the frequency of around 50 MHz. This is one of the causes of weak backscattered echoes from the mesosphere. The authors should consider to include this message in the discussion of Fig. 8b.

**Answer:** Dear reviewer, thank you for your suggestion. We added the message in the revised paper.

**Question (2)**

The authors provide two more papers for justification of SSW. One was made for the tropical region, and the other was for Eastern Siberia. Are these latitudinal locations representative for the Wuhan radar location? What is the scale of SSW?

In the response report, the authors show a more detailed time-altitude evolution of the daily mean zonal winds observed between Jan and Feb. Weakening and reversal in zonal wind can be found. Maybe this can be included in Fig. 12 to convince the readers further.

**Answer:** Dear reviewer, thank you for your suggestion. Although the two locations in the papers are at different latitudes, they can prove the occurrence of the SSW events. Therefore, the two papers have certain reference value for the SSW events. The 2016 Feb SSW is a minor SSW, and 2017 Feb SSW is also a minor SSW. To convince the readers, we added the Fig. S1 of time-altitude evolution of the daily mean zonal winds in the supplement.

**More comments:**

**1.** L347:……around 81 km show better agreement than other heights, except for the meridional wind on 12 January 2016. I think the wind difference in Fig. 11d is large around the height of 81 km.

**Answer:** Dear reviewer, thank you for your suggestion. We modified the sentence in the revised paper.

2. According to my reading, the authors do not provide suitable answers to some questions raised by the reviewer RC2, e.g.,

(1) Question 4: What is the algorithm in frequency domain for removal of the ground clutter?

(2) Question 7: It is ERA-interim mentioned in the question, not EAR radar.

**Answer:** Dear reviewer, the process of the algorithm is removing several points around zero frequency and filling them with interpolated data from neighboring points. Although the algorithm is simple, the effectiveness is satisfactory. About the second question, we added the analysis of correlation coefficient in the revised paper.

**Response to Referee #2**

**Question (1)**

The grammatical or syntactic errors on line 13, 14, 38, 71, 232-233, 316, 367, the reference errors on line 34, 222, 304, and the legend errors of Fig. 9 need be modified.

**Answer:** Dear reviewer, we modified the errors in the revised paper.

**Question (2)**

Please see my comment No.7, asked the authors to perform some statistical analysis on winds derived by the radar and ERA-reanalysis (not EAR radar).

This is the main weakness of the paper. The measurements show large deviations (at some heights) from radiosonde and reanalysis winds. The authors neither show poof for their claims nor provided a valid justification for the discrepancy.

**Answer:** Dear reviewer, we added the analysis of correlation coefficient in the revised paper. After analysis, we believe that 
[revised manuscript text omitted]

---

## Author Response (AR3)

**Response to review comments of amt-2020-17**

Wuhan MST radar: Technical features and Validation of wind observations

Lei Qiao, Gang Chen, Shaodong Zhang, Qi Yao, Wanlin Gong, Mingkun Su,
Feilong Chen, Erxiao Liu, Weifan Zhang, Huangyuan Zeng, Xuesi Cai, Huina Song,
Huan Zhang, Liangliang Zhang

August 20, 2020

Dear Editor:

Please find enclosed the revision of our submission “Wuhan MST radar: Technical features and Validation of wind observations” (ID: amt-2020-17).

We would like to thank you for handling the review process of our paper. We are also indebted to the reviewers for their helpful comments. In this revision, all of the comments raised have been addressed and marked in the revised manuscript. A detailed point-by-point response to the comments is given below.

We appreciate for Editors/Reviewers' warm work earnestly, and hope that the correction will meet with approval. Once again, thank you very much for your comments and suggestions.

Yours sincerely,

Lei Qiao

**Note:** To help legibility of the remainder of this response letter, all the reviewers' comments and questions are written in black color. Our responses and remarks are written in blue color.

**Response to Editor**

**Question**

One point remains questionable, which is related to the correlation of the zonal and meridional winds for ERA-interim and the MST radar. Note that a correlation is large when the dynamic range of the correlated variable is large with respect to the difference in the variable. Note also that the dynamic range of the zonal wind is much larger than that of the meridional wind (Fig. 10). This leaves the question whether the difference in correlation between zonal and meridional wind is related to the different dynamic ranges or to larger differences in the meridional wind (in m/s). The latter is most relevant for the MST radar validation. Please clarify by presenting difference statistics (e.g., SD, RMS, or MAD) for this data set.

**Answer:** Dear editor, thank you for your suggestion. The correlation coefficients of 0.96 (zonal) and 0.7 (meridional) indicate the zonal winds have a more consistent trend, and the root mean square errors (RMSE) of 9.3 (zonal) and 4 (meridional) indicate the meridional winds have a smaller deviation. Therefore, the larger correlation coefficient of the zonal winds may be attributed to the larger dynamic range. We have added the message in the revised paper.

**Wuhan MST radar: Technical features and Validation of wind observations**

Lei Qiao1,2, Gang Chen2, Shaodong Zhang2, Qi Yao3, Wanlin Gong2, Mingkun Su1,
Feilong Chen4, Erxiao Liu1, Weifan Zhang2, Huangyuan Zeng2, Xuesi Cai1, Huina Song1,
5 Huan Zhang1, Liangliang Zhang1

[revised manuscript text omitted]